

# Transformation and aging of biomass burning carbonaceous aerosol over tropical South America from aircraft in-situ measurements during SAMBBA

William T. Morgan[1], James D. Allan[1,2], Stéphane Bauguitte[3], Eoghan Darbyshire[1], Michael J. Flynn[1], James Lee[4], Dantong Liu[1], Ben Johnson[5], Jim Haywood[5,6], Karla M. Longo[7*], Paulo E. Artaxo[8], and Hugh Coe[1]

[1]School of Earth & Environmental Sciences, University of Manchester, Manchester, UK
[2]National Centre for Atmospheric Science, University of Manchester, Manchester, UK
[3]Facility for Airborne Atmospheric Measurements, Cranfield University, UK
[4]Department of Chemistry, University of York, York, UK
[5]Met Office, Exeter, UK
[6]College of Engineering, Mathematics and Physical Sciences, University of Exeter, UK
[7]National Institute for Space Research (INPE), Sao Jose dos Campos, Brazil
[8]Physics Institute, University of Sao Paulo, Sao Paulo, Brazil
[*]Now at NASA Goddard Space Flight Center and USRA/GESTAR, Greenbelt, MD, USA

**Abstract.**

We present a range of airborne in-situ observations of biomass burning carbonaceous aerosol over tropical South America, including a case study of a large tropical forest wildfire and a series of regional survey flights across the Brazilian Amazon and Cerrado. The study forms part of the South American Biomass Burning Analysis (SAMBBA) Project, which was conducted during September and October 2012. We find limited evidence for net increases in aerosol mass through atmospheric aging combined with substantial changes in the chemical properties of organic aerosol (OA). Oxidation of the OA increases significantly and rapidly on the scale of 2.5-3 hours based on our case study analysis and is consistent with secondary organic aerosol production. The observations of limited net enhancement in OA coupled with such changes in chemical composition, imply that evaporation of OA is also occurring to balance these changes. We observe significant coatings on black carbon particles at source, but with limited changes with aging in both particle core size and coating thickness.

We quantify variability in the ratio of OA to carbon monoxide across our study as a key parameter representing both initial fire conditions and an indicator of net aerosol production with atmospheric aging. We observe ratios of 0.075-0.13 $\mu g\, sm^{-3}\, ppbv^{-1}$ in the west of our study region over the Amazon tropical forest in air masses less influenced by precipitation and a value of 0.095 $\mu g\, sm^{-3}\, ppbv^{-1}$ over the Cerrado environment in the east. Such values are consistent with emission factors used by numerical models to represent biomass burning OA emissions. Black carbon particle core sizes typically range from 250-290 nm, while coating thicknesses range from 40-110 nm in air masses less influenced by precipitation. The primary driver of the variability we observe appears to be related to changes at the initial fire source. A key lesson from our study is that the complex nature of the regional aerosol and its drivers precludes aggregating our observations as a function of atmospheric aging due to the many conflating and competing factors present.



20 Our study explores and quantifies key uncertainties in the evolution of biomass burning aerosol at both near-field and regional scales. Our results suggest that the initial conditions of the fire are the primary driver of carbonaceous aerosol physical and chemical properties over tropical South America, aside from significant oxidation of OA during atmospheric aging. Such findings imply that uncertainties in the magnitude of the aerosol burden and its impact on weather, climate, health and natural ecosystems most likely lie in quantifying emission sources, alongside atmospheric dispersion, transport and removal rather 25 than chemical enhancements in mass.

## 1 Introduction

Biomass burning represents a significant source of aerosol particles on the global scale and thus has a substantial impact on the Earth System. At the regional level, where large-scale and seasonal burning practices are conducted annually, significant anthropogenic perturbations may occur. The Brazilian Amazon Rainforest and Cerrado are such regions, where an annual 30 burning season typically running from August to October results in the build-up of a large atmospheric aerosol burden that can affect climate (e.g. Andreae, 2004), weather (e.g. Kolusu et al., 2015), human health (e.g. Reddington et al., 2015) and the regional ecosystem (e.g. Crutzen and Andreae, 1990; Pacifico et al., 2015). The co-emission of large amounts of both gas and particle phase pollutants from biomass burning complicates assessments of its impact as they undergo physical and chemical processes downwind.

35 Biomass burning is the largest source of black carbon (BC) particles when considering the global scale (Bond et al., 2013). One of the key complicating factors in relation to BC is the co-emission and subsequent mixing with other chemical components (e.g. Bond et al., 2013); aerosol components that predominantly scatter solar radiation can increase the absorption by BC (e.g. Bond et al., 2006) with the mixing state of the particles being vital (Liu et al., 2017). Furthermore, the relative abundance of BC compared to other (scattering) components in the aerosol will strongly govern the radiative impact from a specific 40 emission source (Haywood and Shine, 1995). Koren et al. (2004, 2008) illustrated that absorption by biomass burning smoke over the Amazon can inhibit cloud formation or even induce so-called cloud "burn-off". Darbyshire et al. (2018) presented average vertical profiles of BC coating thickness across Brazil with significantly thinner coatings in the east than the west of the Amazon Basin.

As well as BC, biomass burning produces abundant emissions of organic carbon species in the gas and particle phase (e.g. 5 van der Werf et al., 2017). As well as direct emissions of primary organic aerosol (POA), biomass burning produces significant emissions of semi-volatile or intermediate-volatility non-methane organic carbon species that may potentially play a role in secondary organic aerosol (SOA) formation (Yokelson et al., 2013b; Stockwell et al., 2015; Shrivastava et al., 2017). A number of field studies have attempted to quantify changes in organic aerosol (OA) production downwind of fire sources, which is a key parameter for global models (Shrivastava et al., 2017). Many such studies have used carbon monoxide as a largely inert 10 chemical tracer on the scale of anticipated changes in biomass burning emissions through aging and then compared this with OA mass concentrations downwind to investigate any net changes. Previous airborne studies have reported both an increase in OA mass (DeCarlo et al., 2008; Yokelson et al., 2009) and a decrease (Hobbs, 2003; Akagi et al., 2012; Jolleys et al., 2012), with



the majority of studies reporting no detectable net addition of OA mass (Capes et al., 2008; Cubison et al., 2011; Hecobian et al., 2011; Forrister et al., 2015; Jolleys et al., 2015; May et al., 2015; Liu et al., 2016). Ground-based studies have also

reported conflicting results with Lee et al. (2008) reporting enhancements in plumes from two prescribed fires in the US, Zhou et al. (2017) reporting no net enhancement for wildfire emissions in the US, while southern African biomass burning plumes have shown net enhancements with aging (Vakkari et al., 2014, 2018). One consistent trend within such studies is the increasing oxidation of the organic aerosol with atmospheric aging (Shrivastava et al., 2017, and references therein). Such observations imply that SOA is being produced and a number of studies (e.g. Capes et al., 2008; Cubison et al., 2011; Jolleys et al., 2012,

2015; May et al., 2015; Zhou et al., 2017) have hypothesised that this is frequently balanced by the dilution and evaporation of POA mass to explain the limited net changes observed in the field. Furthermore, in studies where net enhancements are observed, the magnitude of the enhancements are smaller than those observed in urban or biogenic emissions (Shrivastava et al., 2017).

The goal of this paper is to characterise the transformation and aging of biomass burning aerosol over the Amazon Basin

with a focus on the carbonaceous component. The experimental study was conducted during the airborne component of the South American Biomass Burning Analysis (SAMBBA) during September/October 2012. SAMBBA aimed to investigate the impact of biomass burning in the region on the Earth System and on human health. SAMBBA represented the first airborne deployment of an Aerosol Mass Spectrometer and Single Particle Soot Photometer in Brazil, providing previously unobtainable characterisation of non-refractory aerosol species and BC-containing particles and their microphysical properties with high

time-resolution and sensitivity. Such measurements allow us to investigate the composition of OA and mixing state of BC as a function of atmospheric aging for a case study analysis of a large fire plume as well as regional-scale measurements across the Amazon Basin. The analysis presented here serves as the bridge between near-source fire emissions characterised by Hodgson et al. (2018) and a regional-scale synthesis by Darbyshire et al. (2018). Compared to previous and subsequent biomass burning seasons, 2012 was a relatively 'normal' year when compared to the past decade (Darbyshire et al., 2018), which has seen

reduced deforestation compared to the historical record albeit with less significant reductions in fire count (e.g. Aragão et al., 2018). As such, the observations presented here provide a characterisation of biomass burning aerosol under relatively 'typical' conditions.

## 2  Method

### 2.1  Instrumentation

All measurements presented here were conducted on the UK Facility for Airborne Atmospheric Measurement (FAAM) British Aerospace 146 (BAe-146) Atmospheric Research Aircraft. As a whole, SAMBBA (Facility for Airborne Atmospheric Measurements, Natural Environment Research Council, Met Office and Coe, 2014) was composed of eighteen science flights conducted between 14 September and 3 October 2012. For the purposes of our regional analysis, we investigate nine of these flights, which focussed on boundary-layer sampling of the regional biomass burning haze (see Darbyshire et al. (2018) for a



broader discussion and context for SAMBBA). The flights and their operating regions are summarised in Table 1. The primary base of operations was Porto Velho in Rondônia state.

The aerosol instrumentation used in this study sampled via Rosemount inlets (Foltescu et al., 1995). These inlets have been shown to be satisfactory for sub-micron aerosol measurements (Trembath et al., 2012), which is typical of the SAMBBA dataset as a whole based on size distribution measurements. Nafion driers were used to dry the aerosol sample, which in combination

with ram heating as the sampled air enters the aircraft and decelerates, reduced the measured sample relative humidity to a range from 20-60%. Losses associated with the nafion driers represent an additional uncertainty in our measurements, which we do not account for. All concentrations are reported at standard temperature and pressure (STP, 273.15 K and 1013.25 hPa respectively) and are denoted with an 's' in their unit where appropriate.

A Droplet Measurement Technologies (DMT, Boulder, CO, USA) Single Particle Soot Photometer (SP2, Stephens et al.,

2003) measured BC physical properties during this study. The SP2 measures what is generally referred to as refractory BC or rBC, as defined by Petzold et al. (2013). Details of the instrument setup on the FAAM research aircraft and data processing relevant to this study have been detailed elsewhere (Liu et al., 2010; McMeeking et al., 2010). The incandescence signal of the instrument was calibrated using Aquadag BC particle standards (Aqueous Deflocculated Acheson Graphite, manufactured by Acheson Inc., USA) to calculate BC mass using a scaling factor of 0.75 (Baumgardner et al., 2012) to account for differences

between the reference BC standard and ambient BC. As in previous studies, we assume a 30% uncertainty in the SP2 BC mass (e.g. McMeeking et al., 2010, 2012). Following the method presented by Liu et al. (2014) and Taylor et al. (2015) a rBC spherical equivalent core diameter, $D_c$, is derived and related to the particle diameter, $D_p$, which represents both the rBC core and its associated coating. By assuming a full concentric encapsulation of the spherical core, the coating thickness of single rBC particles are estimated using a core refractive index of 2.26-1.26$i$ and coating refractive index of 1.50+0$i$.

An Aerodyne Research Inc. (ARI, Billerica, MA, USA) compact Time-of-Flight Aerosol Mass Spectrometer (AMS, Drewnick et al., 2005; Canagaratna et al., 2007) measured non-refractory OA and inorganic component mass (sulfate, nitrate, chloride and ammonium). In terms of OA, the AMS measures all refractory organic matter (OM) rather than just organic carbon (OC) and we thus refer to OM when discussing the AMS data specifically. Details on the instrument setup on the FAAM research aircraft and calibration protocols have been detailed elsewhere (Crosier et al., 2007; Morgan et al., 2009, 2010). Measured

mass concentrations are subject to an uncertainty of approximately 30% (Bahreini et al., 2009). In addition to the standard operating procedure of the AMS, we collected data at 1 s time resolution during discrete fire plume sampling by employing the 'fast mass spectrum' mode of the instrument (Kimmel et al., 2011).

Additional information regarding quality assurance procedures for the SP2 and AMS during SAMBBA can be found in Hodgson et al. (2018).

Carbon monoxide mixing ratios were measured using a vacuum ultraviolet (VUV) fast fluorescence CO analyser, with measurement uncertainties of approximately 2% (Hopkins et al., 2006; O'Shea et al., 2013).





## 2.2 Background concentration calculations

Excess mixing ratios and concentrations of individual species $x$, denoted as $\Delta x$, are necessary in order to investigate chemically-driven changes as well as other processes such as dilution and wet-removal. For the following regional-scale analysis, we use the fifth percentile for each species during a straight-and-level-run (SLR) as the ambient background values of species $x$ to determine $\Delta x$. For the case-study analysis, we identified the smoke plumes manually based on the time series of CO, OA and rBC and then determined the ambient background values while sampling outside the smoke plumes for each SLR using the same method as the regional-scale analysis. Tropospheric mixing can lead to changes in the background air composition, which can lead in uncertainties in the determination of $\Delta x$ (Yokelson et al., 2013a); our sampling and method aims to mitigate for such changes as our SLRs are relatively short (10-30 minutes) and within the atmospheric boundary layer, typically sampling a fairly homogenous haze burden over a single SLR or flight. This limits large changes in mixing plus we manually inspect our time series and background values to identify clear shifts due to changing air masses e.g. large-scale spatial gradients (B734) or wet-scavening (B739) and recalculate background values over shorter flight segments if necessary. For the case-study analysis, our measurements are very close to source and we observed constant background concentrations throughout our plume intercepts. As a result, we expect uncertainties in the determination of $\Delta x$ to be small.

## 3 Tropical forest fire case study

The following section presents a case study of a tropical forest fire sampled on flight B737 on 20 September 2012 in Rondônia state. Take-off was at 14:45 UTC (10:45 local-time), lasting 3 hours 45 minutes. The state is characterised by tropical moist broadleaf forest, as well as extensive deforestation. A large smouldering tropical forest fire was sampled and is shown in Fig. alongside the flight track of the aircraft during the low-level sampling of the fire and smoke plume. Hodgson et al. (2018) reported near-source measurements of the fire, concluding that the fire was likely natural in origin as it was located well away from deforestation areas and was in a national park many kilometres from the nearest road. They reported a modified combustion efficiency (MCE, Ward and Radke, 1993) of $0.79 \pm 0.02$, which is relatively low compared to typical deforestation fires, Ferek et al. (e.g. 1998) reported a value of 0.87 for such fires in Brazil. The fire was characterised by substantial emissions of carbon monoxide and very low emissions of rBC compared to prior literature on Brazilian deforestation fires and global tropical forest fires, as well as being largely composed of OM (97.1% of the sub-micron mass).

The in-situ plume sampling was conducted as a sequence of cross-wind intercepts downwind of the fire at approximately 2500 m above sea level, an along-plume SLR at the same altitude and a series of overpass intercepts directly above the fire at an altitude of approximately 1800 m above sea level. As the fire was located on a 900 m high plateau, the near-source sampling was conducted approximately 900 m above ground-level. These flight sections are illustrated in Figs. and .

We do not have a quantitative estimate of fire size for the B737 case study, although based on MODIS hotspot data corresponding to our sample location, the maximum fire size is approximately 5 km$^2$ and was likely smaller during our sampling. Furthermore, based on the velocity of the aircraft and the width of cross-plume intercepts, we estimate that the plume was approximately 3-4 km wide in the near-field before expanding to approximately 21 km when 56 km downwind of the fire. The



atmospheric stability profile tends towards instability in the flight conditions during our case study, with absolutely unstable air below approximately 2 km where the lapse rate is 10.9 K km$^{-1}$ and conditionally unstable air from 2-3 km where the lapse rate is 7.44 K km$^{-1}$.

The time series of gas and particle-phase species shown in Fig. illustrates the significant enhancements in their concentrations during plume intercepts, as well as the gradual increase in their concentrations as the aircraft approached the fire. CO mixing

ratios were in excess of 5000 ppb on the approach to the fire, climbing to over 15000 ppb when directly above it during the overpass intercepts. OM mass concentrations reached almost 800 µgsm$^{-3}$ on the along-plume SLR and over 3500 µgsm$^{-3}$ during the above-fire intercepts.

Measurements from the along-plume SLR are shown in Fig. relative to the distance from the fire, as well as the approximate age of the plume at the point of sampling. The distance from the fire is calculated as the great circle distance of the aircraft

from the latitude and longitude of the fire (approximately 11.0° S, 63.6° W). The age of the plume is calculated using the average wind speed, which was 6.2 ± 1.7 ms$^{-1}$. We note there was a small gradient in wind speed along the length of the plume of 0.02 ms$^{-1}$km$^{-1}$, corresponding to an average increase of 1.2 ms$^{-1}$ along the length of the plume. We omit this from our calculations due to the variability in wind speed also observed, while noting that the latter plume ages reported are potentially biased towards higher values. The plume extended approximately 65 km downwind based on the concentrations

reaching regional background values, which equates to approximately 3 hours in terms of plume age.

The evolution of ΔOM:ΔCO ratio along the length of the plume indicates negligible net change in OA mass downwind of the fire, with the ratio exhibiting a small net decline over the course of the measurements and a low correlation coefficient of -0.16. Compared to the near-source measurements directly above the fire, the ΔOM:ΔCO ratio is slightly enhanced in the near-field, although the variability in the above-fire ratio is large relative to the difference. Small net enhancements in sulfate and

nitrate relative to CO are observed along the length of the plume, with correlation coefficients of 0.36 and 0.48 respectively. In addition, the along-plume measurements are enhanced relative to the above-fire intercepts. The relative intensity of the organic signal at m/z 44 to the total organic mass, which corresponds to the $CO_2^+$ ion and is denoted as $f_{44}$, is used as an indicator for the level of oxidation of the organic aerosol. We observed a substantial increase in $f_{44}$ along the length of the plume from approximately 0.05 to 0.15, with a correlation coefficient of 0.90. This equates to an increase in O:C of approximately 0.29

to 0.73 over the course of the sampling, which is calculated using the equation from Canagaratna et al. (2015) . We observe a small reduction in coating thickness of the rBC-containing particles of -4.9 ± 1.4 nm hr$^{-1}$ with a correlation coefficient of -0.28. The coating thickness observed during the along-plume sampling is lower than that from the above-fire intercepts, although the variability is large in the latter. We do not observe any change in the ΔrBC:ΔCO ratio along the length of the plume sampling, as well as similar ratios compared to the above-fire intercepts. We observe a minor absolute decrease in rBC

core diameter along the plume (-4.4 ± 2.3 nm hr$^{-1}$) , with a correlation coefficient of -0.16, coupled with similar diameters observed on the above-fire intercepts.

Fig. shows examples of average OM mass spectra at different stages of the plume's evolution, with approximate one order of magnitude decreases in concentration. Consistent with the $f_{44}$ evolution above, the organic aerosol becomes increasingly oxidised downwind of the fire and closely resembles the background regional aerosol after approximately 2.5 hours of aging.



Above the fire, $m/z$ 43 dominates (7.6%) corresponding to CH and CHO ions, with further enhancements from other hydrocarbon peaks, especially $m/z$ 29, 41, 55 and 57. $m/z$ 60, which is associated with levoglucosan and other anhydrous sugars, is also elevated (3.9% of the organic signal); levoglucosan is often reported as a tracer for biomass burning aerosol. The mean absolute difference and correlation between the mass spectrum and the background is 0.0023 and 0.66 respectively, illustrating their similarity. Approximately 1 hour downwind, a similar pattern is observed but now with increased signal at $m/z$ 44 (6.2%), while maintaining the signal at $m/z$ 43 (7.5%) and reduced signal at the hydrocarbon peaks noted above. The contribution of $m/z$ 60 has reduced to 2.7% at this point. The mean absolute difference and correlation between the mass spectrum and the background is 0.0016 and 0.87 respectively, illustrating their similarity. After 2.5 hours, the contribution from the hydrocarbon peaks has reduced substantially and $m/z$ 44 dominates the organic mass spectrum (13.3%). The contribution of $m/z$ 44 increases further in the background organic aerosol (15.7%). The mean absolute difference and correlation between the mass spectrum and the background is 0.00064 and 0.99 respectively, illustrating their similarity. The contribution of $m/z$ 60 has diminished further after 2.5 hours (1.0%), while being close to zero in the background (0.4%).

The evolution in the organic mass spectra is further illustrated in Fig. , where $f_{44}$ is compared with $f_{43}$ and $f_{60}$ for the plume run and regional background aerosol. Relative to the increase in $f_{44}$ along the plume and its eventually comparable magnitude to the regional background, $f_{43}$ initially decreases within approximately the first 45 minutes of the plume's evolution, before a partial increase and stable magnitude up to the 2 hour mark. $f_{43}$ then increases over the rest of the plume run until reaching the regional background value. The points fall within the $f_{44}$ and $f_{43}$ 'triangle' space reported by prior studies focussed on organic aerosol less influenced by biomass burning (Morgan et al., 2010; Ng et al., 2010). Comparing $f_{44}$ and $f_{60}$ illustrates a gradual reduction in $f_{60}$ as the plume ages and becomes increasingly oxidised downwind, falling within the space reported by Cubison et al. (2011) and Lack et al. (2013) who reported a similar linear progression.

## 4 Regional biomass burning haze analysis

The following section examines regional biomass burning flights during SAMBBA to investigate the aging and evolution of the carbonaceous aerosol on the regional scale. Following the case study in Section 3, we relate the evolution of the regional OA based on changes in $f_{44}$ as an indicator of the age of the biomass burning smoke sampled. We couple this with the ratio of rBC to CO as an indicator of the air mass history based on the assumption that both are relatively inert tracers that are strongly controlled by the initial conditions at source; the ratio also provides an indication for the influence of precipitation, which would reduce the rBC mass concentration to a larger extent than CO. Including this ratio in our analysis framework provides a means of isolating net changes in OA mass concentration during aging from changes driven by air mass history.

We focus on boundary layer regional haze, which is determined based on the procedure outlined in Darbyshire et al. (2018) where near-source plumes were identified based on a series of threshold concentrations for multiple pollutants and then flagged separately. This allows us to exclude such plumes from the wider regional haze that we are interested in here.

Fig. 6 illustrates the geographical scope of the flight campaign, with the majority of flights sampling within Rondônia state, as well as western Mato Grosso and Tocantins. Deforestation and degradation fires in tropical forest environments are





the primary source within Rondônia and Mato Grosso, while Cerrado fires dominate in Tocantins. As well as the geographical regions identified, the analysis can be split into distinct meteorological phases following Brito et al. (2014) and Darbyshire et al. (2018). From 14-22 September (flights B731-B737), relatively dry conditions were prevalent and were characteristic of the 'dry season'. Following this period, the monsoonal transition was being established and was characterised by increased precipitation

across the western and southern Amazon Basin. The number of fires was consequently reduced during this transition phase. Further details and analysis of the meteorological fields are available in Darbyshire et al. (2018).

Fig. 7 summarises the aerosol chemical composition for each flight using data from SLRs in boundary layer regional haze. Flight B731 in Rondônia state was the most polluted with total sub-micron mass concentrations of 46 $\mu$gsm$^{-3}$. Lower average concentrations are observed across the remainder of the flights, with B734 in Rondônia and B742 in Tocantins being the next

greatest in terms of total sub-micron mass concentrations of 18 and 17 $\mu$gsm$^{-3}$ respectively. OM dominates the sub-micron chemical composition, ranging from 75% of the total on flight B746 to 86% of the total on flight B739. Sulphate mass fractions ranged from 3.2-9.7% and are typically larger than nitrate mass fractions, which generally fell between 1.7-3.9% with flight B740 as an outlier with 7.7%. Chloride mass fractions were low, ranging from 0.2-0.6%. Based on ion balance calculations of the inorganic aerosol species, the aerosol was typically neutralised. rBC mass concentrations varied from 2.0-6.1% on flights

within Rondônia state, with the largest concentrations on flights B731 and B734. Note the data coverage for the SP2 was more limited on flight B731 than the other flights. Average rBC mass concentrations (1.5 $\mu$gsm$^{-3}$) were greatest on flight B742 in Tocantins, contributing 8.7% of the total sub-micron mass concentration. The largest contribution by rBC was 12.4% on flight B746, which sampled within both western Mato Grosso and Rondônia.

Fig. 8 illustrates the relationship between excess concentrations of OM and CO for each flight. For additional context, the

points are coloured by the ratio of rBC to CO. Broadly speaking, there is a strong linear relationship between OM and CO, with correlation coefficients ranging from 0.57-0.98. However, the ratio varies both between and within flights. Variability within individual flights e.g. B734, B737, B739 and B746 is coincident with differences in the ratio of rBC to CO, which likely reflects differences in air mass history across the region(s) sampled on the flight.

Fig. 9 shows the relationship between the ratio of OM and CO compared with $f_{44}$ to examine whether the ratio changes

with variability in OA oxidation and aging. Within an individual flight, we observe a limited relationship between the ratio and the level of oxidation of the OA, with predominantly low correlation coefficients ranging from -0.09-0.09, except for B734 (0.51) and B740 (-0.26). On some flights (B731, B734, B745) there are enhancements for $f_{44}$ greater than approximately 0.16, although in the case of B734 and B745 they appear at least partially related to a change in the ratio of rBC and CO; the limited SP2 data coverage for B731 precludes analysis, although we note that there is also a reduction in the ratio at greater $f_{44}$ values. Comparing across all flights, changes in the ratio of OM and CO compared to the level of oxidation appear related to changes in the ratio of rBC and CO, which suggests a link with air mass history and any perceived change in net condensation or

5   evaporation of OA.

Figs. 10 and 11 examine $f_{44}$ compared with f$_{43}$ and f$_{60}$ in a similar manner to Fig. in Section 3. The majority of the regional haze data shown in Fig. 10 fall within the 'triangle' space reported by prior studies focussed on OA less influenced by biomass burning (Morgan et al., 2010; Ng et al., 2010). Some flights display a broader range of values suggesting that the flights sampled





a more diverse range of air masses in terms of their chemistry and aging. The behaviour described in Section 3 in relation to the evolution during the early stages of the plume's age are present in flights B731, B742 and B745, which suggests sampling of fresher biomass burning smoke on those flights; on flights B742 and B745, such features are distinct in terms of the ratio of rBC and CO as well. In terms of the $f_{44}$ and $f_{60}$ space shown in Fig. 11, the regional sampling is predominantly confined to lower $f_{60}$ values, as well as displaying the linear tendency noted for the case study in Section 3 and previous work (e.g. Cubison et al., 2011; Lack et al., 2013).

Fig. 12 presents histograms of the median rBC coating thickness across the individual flights, which appears to vary appreciably from flight-to-flight, as well as within some individual flights. We found no clear and consistent relationship between coating thickness and $f_{44}$ across the dataset. The broad bimodal-like structure in coating thickness in flight B737 could be linked with differences in the ratio of rBC and CO, with the thicker coatings of 80-100 nm associated with a larger ratio; conversely, the thinner coatings of less than 60 nm are coincident with the smaller ratios observed. However, there was no clear pattern in this linkage in flights B740 and B744, which also had a bimodal-like structure in coating thickness. Regional-scale variability in rBC coating thickness appears to be predominantly driven by fire-source and/or air mass differences, rather than aging of the aerosol population after emission. We also observed no clear link between the physical size of the rBC core and $f_{44}$, with geometric mean mass diameters typically between 250-290 nm.

## 5   Discussion

Whether considering the case study analysis in Section 3 or regional analysis in Section 4, we observe either limited or no net enhancement in the ratio of OM to CO. However, we do observe substantial increases in $f_{44}$, which is interpreted as an indicator for the O:C content of the OA. Such a trend with atmospheric aging is consistent with SOA being produced downwind of source following dilution but that this is approximately balanced by the loss of POA emitted at source. A number of studies have observed such features in other biomass burning environments and hypothesised such a process (e.g. Cubison et al., 2011; Jolleys et al., 2012, 2015; May et al., 2015; Zhou et al., 2017). An additional feature of our observations of the single plume case study is the apparent plateau in $f_{44}$ approximately 2.5 hours downwind of source; we also observe a similar plateau in net ozone production and reduction in nitrogen dioxide downwind. After reaching this plateau, the level of oxidation is comparable to the regional background with highly similar organic mass spectra as well. Such observations suggest that whatever chemical process drives the aging of OA, it is relatively fast under the environmental conditions of our measurements. We note that our case study is likely of a natural fire and that it is highly-smouldering compared to previous fires sampled in Brazil (Hodgson et al., 2018), which may have a bearing on our observations. While we cannot directly ascertain the regional evolution of $f_{44}$ with atmospheric aging, our results imply similar phenomena are present as $f_{44}$ increases with decreasing OA concentrations as well as reaching a defined 'end-point' at approximately 0.20 in the $f_{44}$ vs $f_{43}$ and $f_{60}$ spaces.

Bian et al. (2017) examined the role of a number of factors that could control SOA production in ambient plumes, including fire area as a driver of dilution rate, mass emission flux and atmospheric stability. Based on Bian et al. (2017), our estimated initial fire size and atmospheric stability conditions would lead to some evaporation of OA in the near-field but with significant





SOA production downwind that could balance the initial loss of particulate, which would be consistent with our observations of limited net enhancement in OA. Based on thermodynamic analysis of the SAMBBA experiment by Darbyshire et al. (2018), our regional sampling was typically conducted in unstable air, which is consistent with our observations of net OA production.

45    Aqueous processing of biomass burning emissions has been identified as a potential source of SOA (e.g. Gilardoni et al., 2016; Tomaz et al., 2018), which is likely an important component in Amazonia. However, we are not in a position to assess the role of such processes through our observations. More oxidised OA is thought to be more hygroscopic (e.g. Jimenez et al., 2009), so such processes have implications for the lifetime and radiative impact of biomass burning smoke.

Our regional analysis illustrates the importance of evaluating changes in the aging of regional OA within a framework
50    that also accounts for differences in air mass history. Differences in vegetation, fire dynamics and environmental conditions can result in significant diversity in the absolute and relative emissions of different pollutants from biomass burning that will manifest in the regional aerosol burden. We also observe significant variability from flight-to-flight, even within the same region that is likely a consequence of differing meteorological conditions e.g. the influence of precipitation, as well as changes in fire dynamics. Were we to interpret our observed changes in the ratio of OA to CO as a function of $f_{44}$, we would see enhancements of 2-3 in some instances that are most likely driven by differences in air mass history and fire dynamics rather than chemical processing.

Our results indicate that uncertainties in the magnitude of the aerosol burden most likely lie in quantifying emission sources, alongside atmospheric dispersion, transport and removal rather than chemical enhancements in mass. Across our study, $\Delta$OM:$\Delta$CO ratios range from 0.029-0.13 $\mu\mathrm{g\,sm^{-3}\,ppbv^{-1}}$ in the west of the Amazon Basin, with a value of 0.095 $\mu\mathrm{g\,sm^{-3}\,ppbv^{-1}}$ in the Cerrado-environment sampled on flight B742. Numerical models that attempt to represent the magnitude of the atmospheric aerosol burden typically use fixed emission factors for distinct ecosystems e.g. those used in the
fourth version of the Global Fire Emissions Database (GFED4, van der Werf et al., 2017) report $\Delta$OM:$\Delta$CO ratios of 0.10 $\mu\mathrm{g\,sm^{-3}\,ppbv^{-1}}$ and 0.09 $\mu\mathrm{g\,sm^{-3}\,ppbv^{-1}}$ for tropical forest and savannah fires respectively (assuming an OM:OC ratio of 1.6 for biomass burning OA following Yokelson et al. (2009) and Akagi et al. (2012)). In the western Amazon Basin, our central estimate for the $\Delta$OM:$\Delta$CO ratio when considering all flights together is 0.09-0.10 $\mu\mathrm{g\,sm^{-3}\,ppbv^{-1}}$, closely matching the value for tropical forests in GFED4. Based on the local and synoptic scale situations during flights B739 and B746, we suspect
that the lower observed ratios on these flights are a consequence of wet removal; the lower ratio associated with limited rBC in B734 is less clear based on the large-scale synoptic situation, so we do not speculate on a cause in this case. If these three flights with lowered observed $\Delta$OM:$\Delta$CO ratios are excluded from the analysis then our regional values range from 0.075-0.13 $\mu\mathrm{g\,sm^{-3}\,ppbv^{-1}}$ in the west of the Amazon Basin, indicating significant variability compared to any assumed fixed emission factor. However, we note that variability of 25-30% is much lower than the discrepancy reported between measurements and
models quantifying the aerosol burden over tropical South America where factors ranging from 1.5-6 are required to match satellite and ground-based observations of aerosol optical depth (Reddington et al., 2016, and references therein).

In terms of inorganic species, we observe small net enhancements in sulfate and nitrate relative to CO in the case study analysis. We do not observe clear enhancements in nitrate at the regional scale, while we do observe minor absolute increases in sulfate on some flights.



Aside from the comparison between the above-fire intercepts and the along-plume sampling in the case study, we do not observe clear changes in rBC coating thickness with plume age. Our observations indicate that rBC is rapidly coated in the near-field based on our case study, as well as other near-field sampling during SAMBBA, in contrast to many urban sources and environments (e.g. Liu et al., 2017). This is consistent with previous measurements of North American wildfire emissions, which also showed significant coatings on near-field rBC particles (Schwarz et al., 2008; Sedlacek et al., 2012). On the regional

scale, we observe no clear link between coating thickness and $f_{44}$, with variability in coating thickness appearing to be driven by fire-source and/or air mass differences. Prior measurements of biomass burning emissions over Boreal Canada by Taylor et al. (2014) reported that scavenging of rBC via wet deposition preferentially removed the largest and most coated particles. Such processes may explain some of our observed variability in coating thickness on the regional scale, although we observe limited variability in rBC core size across the dataset. Our measurements indicate that the vast majority of rBC-containing

particles within the boundary layer are coated. Such coatings will lead to a lowering of rBC lifetime as rather than being hydrophobic, the rBC containing-particles will be at least mildly hygroscopic, making them more susceptible to cloud-activation and wet removal. Furthermore, such coatings have implications for the radiative impact of black carbon containing particles via enhanced absorption (Liu et al., 2017). The lack of change in coating thickness appears consistent with the limited enhancement in OA, although given that our results imply a process of both evaporation and condensation of OA-related species there

are potentially complex particle dynamics occuring within the rBC-containing fraction that warrant further investigation.

## 6  Conclusions

We observe limited to no enhancement in OA mass production during atmospheric aging of biomass burning over Brazil for both a case study of a likely natural tropical forest fire and regional sampling over the Amazon and Cerrado. Variability in the ratio of OA to CO is predominantly driven by regional differences likely related to changes at the initial fire source, as well as

air mass differences likely as a consequence of wet scavenging of the aerosol. What enhancements we do observe are small in absolute terms compared to regional-scale variability across our study. Such variability at the regional-scale can be significant, with flight-averaged $\Delta OM$:$\Delta CO$ ratios ranging from 0.075-0.13 $\mu g\, sm^{-3}\, ppbv^{-1}$ across our study region in cases where we suspect the influence of precipitation to be minor. We did not observe a systematic difference between the west of our study region and the Cerrado in terms of $\Delta OM$:$\Delta CO$, although we only have one flight in the latter region. While significant, we note

that the scale of the variability is much smaller than typical factors required to match satellite and ground-based observations of aerosol optical depth to numerical model estimates of the aerosol burden. We do observe substantial changes in the chemical composition of OA, with significantly increased oxidation downwind, implying SOA formation that is being balanced by evaporation of OA. During our case study, we observed an increase in O:C of approximately $0.25 \pm 0.09$ $hr^{-1}$, reaching a plateau after approximately 2.5-3.0 hours of atmospheric aging. Such changes may enhance the hygroscopicity of the OA and

given its dominance of the aerosol burden (75-86% of the sub-micron mass in our study), this will have implications for the life cycle and radiative impact of biomass burning aerosol in the region.



We observe limited changes in the microphysical properties of rBC subsequent to emission, with neither significant changes in particle core size or coating thickness. We observe substantial coatings on rBC-containing particles at source. Given the limited changes with aging, our results suggest that any absorption enhancements will be dictated by the initial conditions in

the near-field and precipitation-influences, rather than aging, although we have not investigated particle morphology changes that may occur. Such coatings likely reduce the lifetime of rBC as they are likely to be at least mildly hygroscopic, especially compared to un-coated hydrophobic rBC particles.

The complex nature of the regional aerosol and its drivers implies that aggregating our observations from the entire study as a function of atmospheric aging is unwise due to the many conflating and competing factors present. The continuing puzzle

over the contrasting observations of the evolution of OA:OC ratios with atmospheric aging remains, although our results appear consistent with the framework presented by Bian et al. (2017). Further detailed quantification of the processes driving these should be further explored in the literature, as well as chamber and ambient studies specifically designed to probe such processes.

Overall, our results suggest that the initial conditions are the biggest driver of carbonaceous aerosol composition and physical

properties in the region, aside from significant oxidation of OA during atmospheric aging. Uncertainties in the magnitude of the aerosol burden and its impact most likely lie in quantifying emission sources, alongside atmospheric dispersion, transport and removal rather than chemical enhancements in mass.

**Data availability**

All raw time series data from the FAAM research aircraft are publically available from the Centre for Environmental Data

Analysis website, where the entire SAMBBA dataset may be accessed. AMS mass spectral features, SP2 size distribution and coating thickness data is available on request. Data masks for categorising flight patterns into plume-sampling and other sampling types (vertical profiles and SLRs) are currently available on request. Active fire data used in the manuscript is available publically from NASA (see acknowledgements for further details).

*Author contributions.*  W. T. Morgan analysed the data and wrote the manuscript. J. D. Allan, E. Darbyshire, J. Lee and D. Liu provided

additional data analysis support, including data processing and quality assurance. S. Bauguitte and J. Lee operated the gas-phase instruments, while J. D. Allan and M. J. Flynn operated the aerosol instruments during the field campaign. B. Johnson, J. M. Haywood, K. M. Longo, P. E. Artaxo and H. Coe led the planning of the field campaign and were co-principal investigators on the SAMBBA project.

*Acknowledgements.*  We would like to acknowledge the substantial efforts of the whole SAMBBA team before, during and after the project. E. Darbyshire was supported by NERC studentship NE/J500057/1 and NE/K500859/1. This work was supported by the NERC SAMBBA

project NE/J010073/1. P. E. Artaxo acknowledges FAPESP (Fundação de Amparo à Pesquisa do Estado de São Paulo) grants 2017-17047-0 and INCT 2014/50848-9. We acknowledge logistical support from the LBA (The Large Scale Biosphere Atmosphere Experiment in



Amazonia) central office, operated by INPA (Instituto Nacional de Pesquisas Espaciais). Active fire data was produced by the University of Maryland and acquired from the online Fire Information for Resource Management System (FIRMS; https://earthdata.nasa.gov/data/near-real-time-data/firms/abouts; specific product: MCD14ML, accessed 4 June 2018).





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



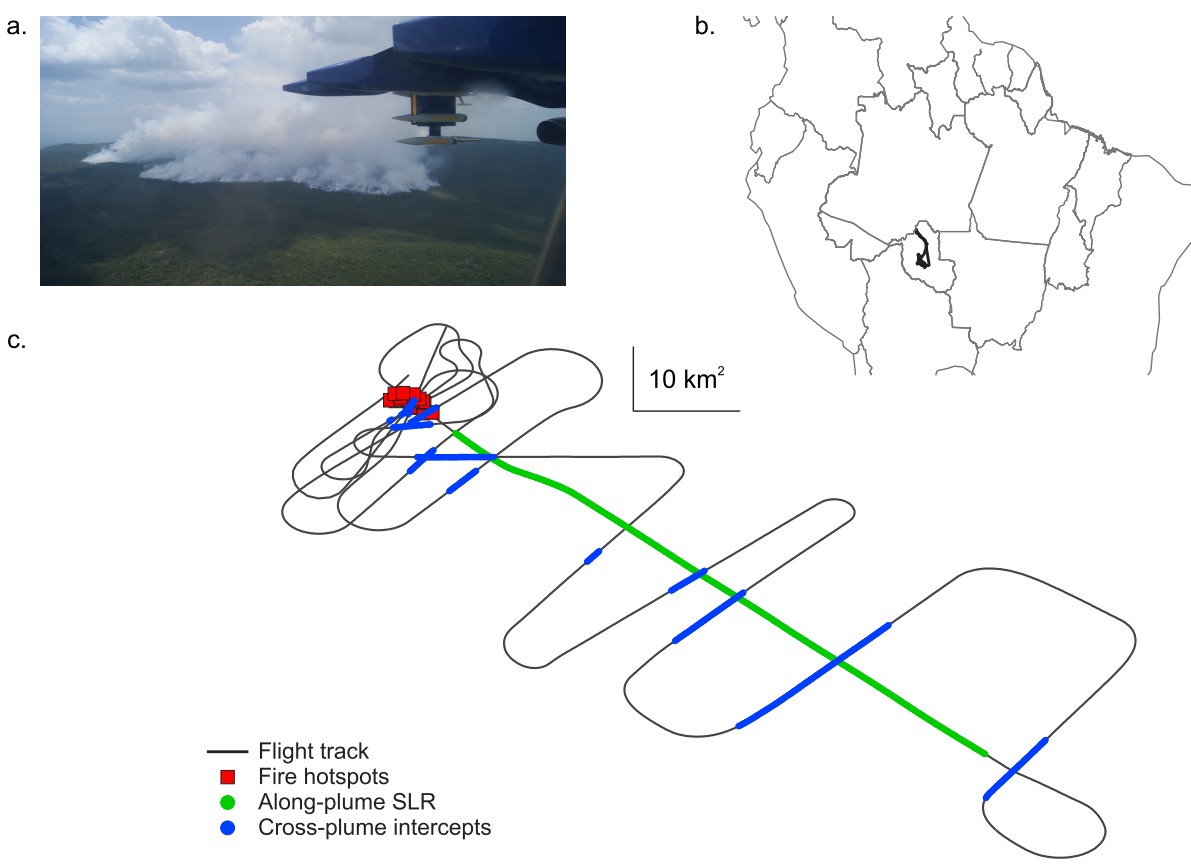

**Figure 1.** Spatial overview of case study analysis for flight B737. (a) Photograph taken from the aircraft of the fire courtesy of William T. Morgan. (b) Flight track of B737 in relation to the wider study region. (c) Low-level flight track of the case study, indicating both cross-plume intercepts and the along-plume straight-and-level run (SLR). Also shown are Moderate Resolution Imaging Spectroradiometer (MODIS) hotspot data from the Terra overpass coincident with our flight sampling. 10 km$^2$ box represents the scale for the flight track.





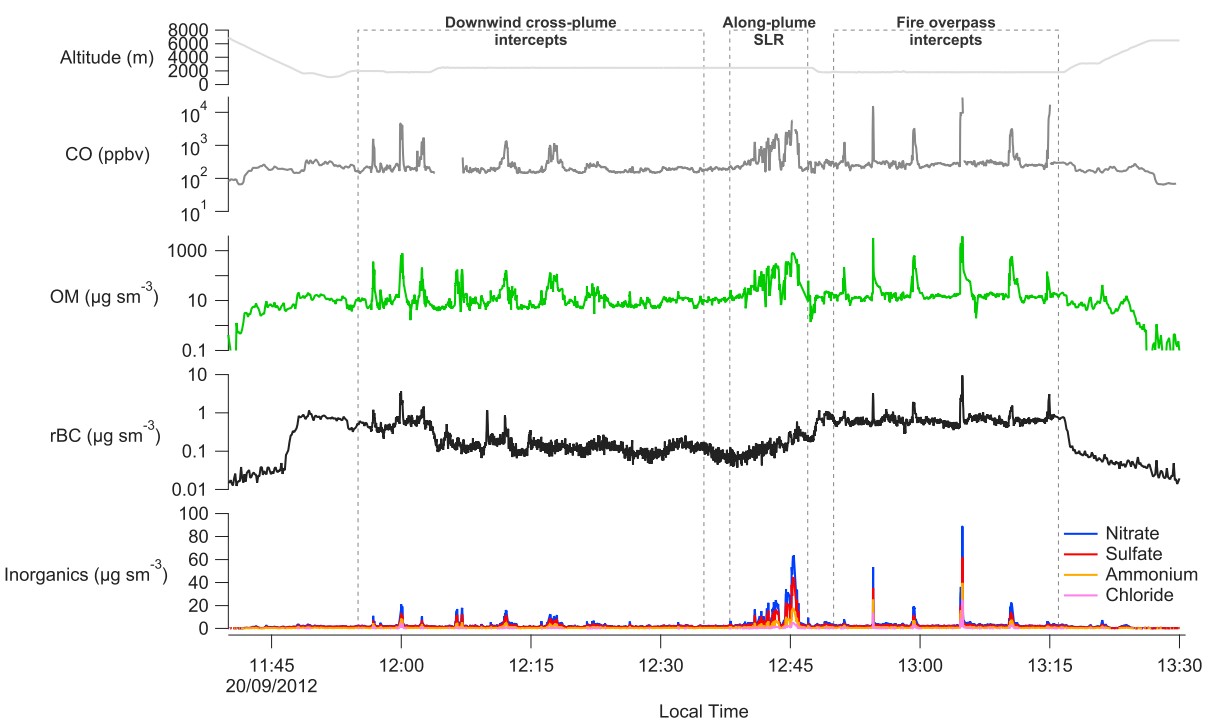

**Figure 2.** Time series of altitude, carbon monoxide (CO), organic matter (OM), refractory black carbon (rBC) and inorganic aerosol components during the case study analysis for flight B737. Downwind cross-plume intercepts, the along-plume straight-and-level run (SLR) and fire overpass intercepts across the plume are indicated by the dashed boxes.



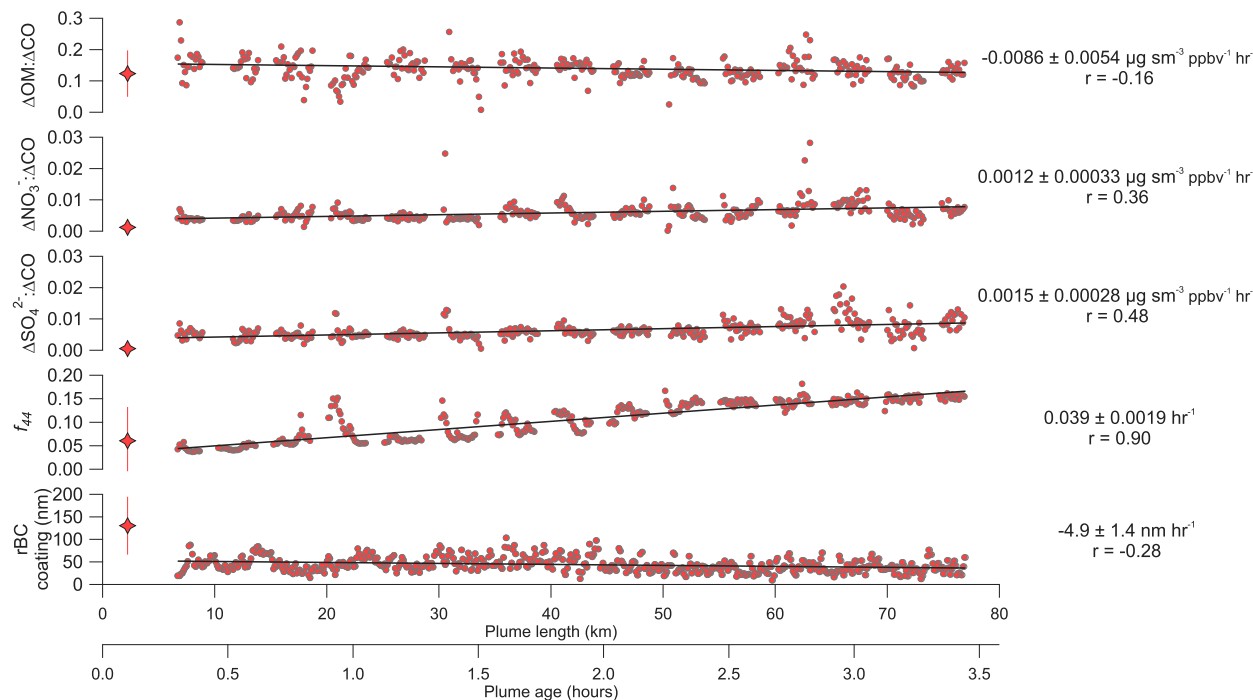

**Figure 3.** Various aerosol chemical and physical parameters as a function of plume length and age for the case study analysis from flight B737. Organic matter (OM), nitrate and sulfate are normalised by carbon monoxide (CO) to account for dilution of the smoke plume downwind. Further details on the calculations are provided in sections2.2 and 3. Red star markers on the left-hand-side of the figure are averaged across from the fire overpass intercepts across the plume with the bars denoting the $2\sigma$ standard deviation range around the mean value. Individual data points are shown as red circles from the along-plume straight-and-level run (SLR) with a linear regression slope included to illustrate any apparent trends. Slopes of the linear regression are given on the right-hand-side of the figure along with their 95% confidence interval. The correlation coefficient, $r$, is also provided.



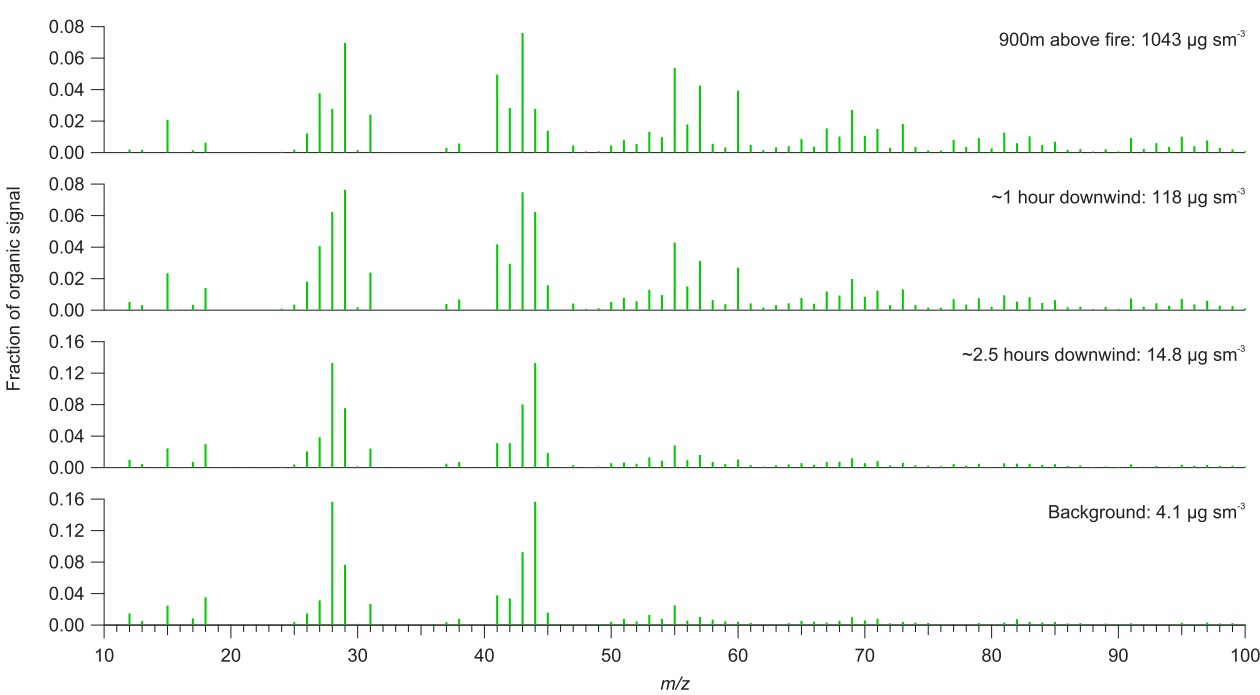

**Figure 4.** Aerosol mass spectrometer organic mass spectra from different segments of the case study analysis from flight B737. The above fire mass spectrum is from sampling directly above the fire during a cross-plume intercept, with the two downwind mass spectra measured during the along-plume straight-and-level run (SLR), while the background mass spectum is in the regional aerosol haze away from the main fire plume study region.





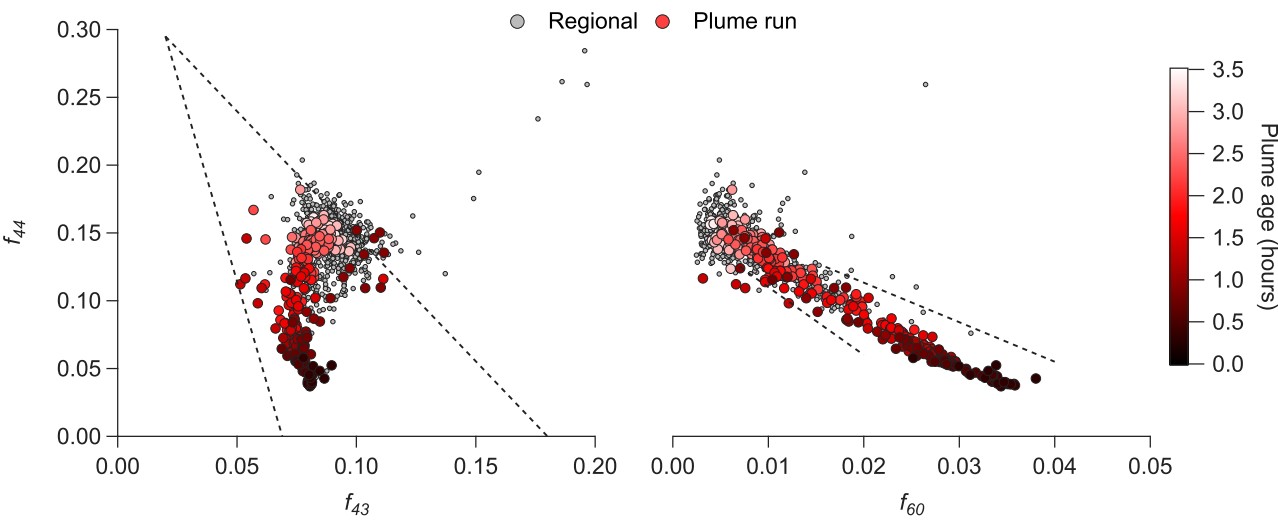

**Figure 5.** Comparison of $f_{44}$ vs $f_{43}$ and $f_{60}$ during the case study analysis for B737, where $f_x$ refers to the fraction of the organic aerosol mass signal at a given mass-to-charge ratio measured by the aerosol mass spectrometer. Also shown is regional haze data during the same flight. Points from the along-plume straight-and-level run (SLR) are coloured according to the approximate plume age. Dashed lines in the $f_{44}$ vs $f_{43}$ show the 'triangle' space reported by prior studies focussed on organic aerosol less influenced by biomass burning (Morgan et al., 2010; Ng et al., 2010). Dashed lines in the $f_{44}$ vs $f_{60}$ are from previous studies on biomass burning organic aerosol aging, with the upper line from Cubison et al. (2011) and the lower line from Lack et al. (2013).



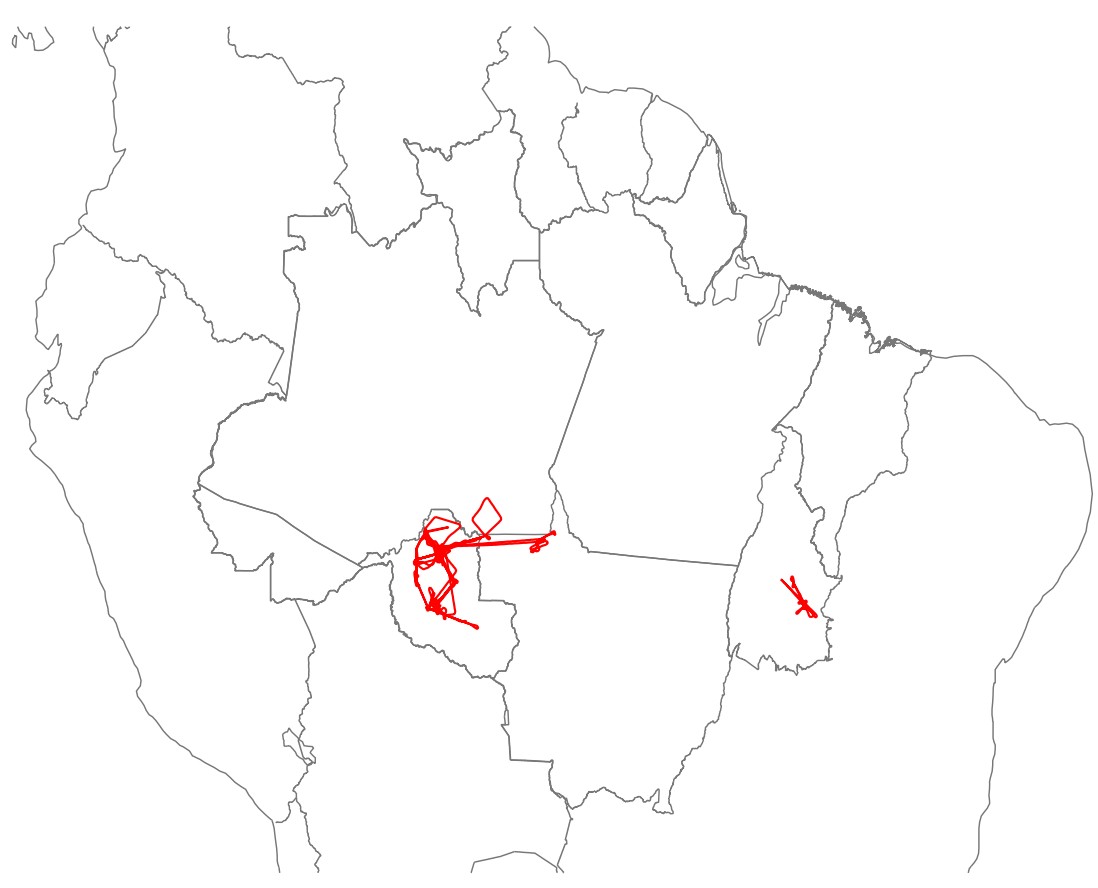

**Figure 6.** Flight tracks of the aircraft during for the flights included our regional analysis. See section 4 and table 1 for further details.




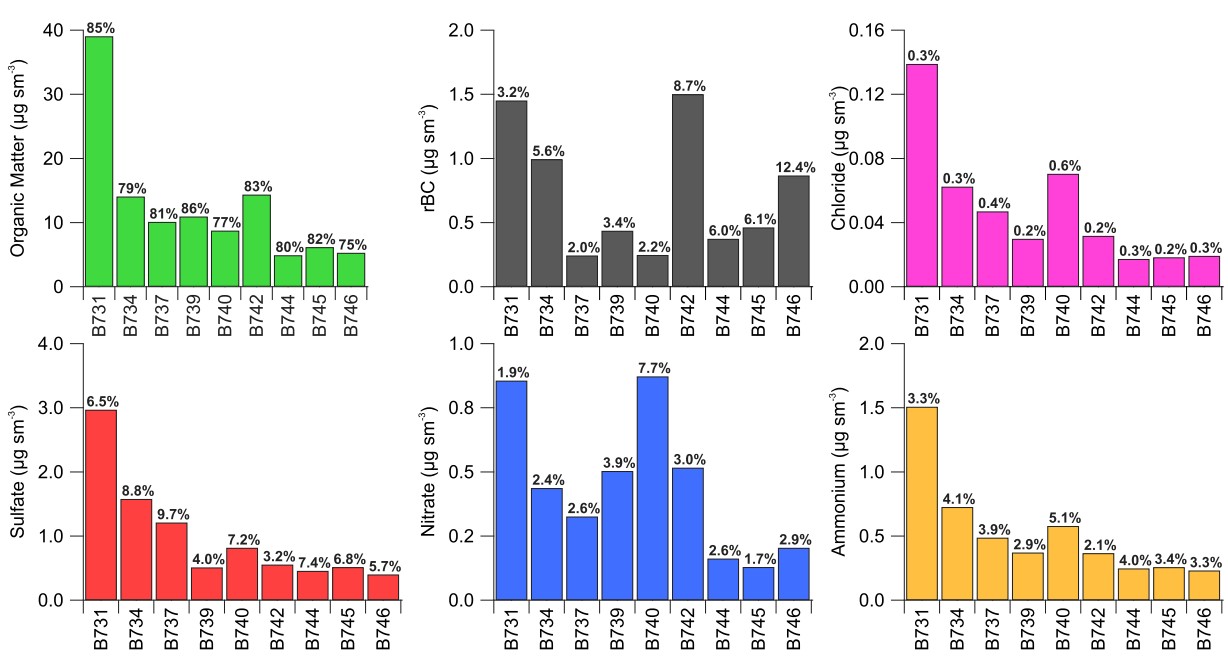

**Figure 7.** Sub-micron aerosol chemical composition overview for the regional analysis with the data split into individual flight operations in the regional boundary layer aerosol haze. Data is from straight-and-level runs (SLRs) only with the bars denoted mean concentrations and the text above each bar providing the mass fraction as a percentage.





**Figure 8.** Comparison of organic matter (OM) with carbon monoxide (CO) across individual flights from the regional analysis. Points are coloured according to the ratio of refractory black carbon (rBC) to carbon monoxide (CO) except for B731 where limited rBC data was available. The black dashed line shows the $0.1\,\mu g\,sm^{-3}\,ppbv^{-1}$ as a consistent baseline for context across all flights, with the solid red lines showing the linear regression for either the whole flight or smaller segments where two lines are shown for a single flight. Red text next to the linear regression lines are the slope of the line-of-best fit in $\mu g\,sm^{-3}\,ppbv^{-1}$.





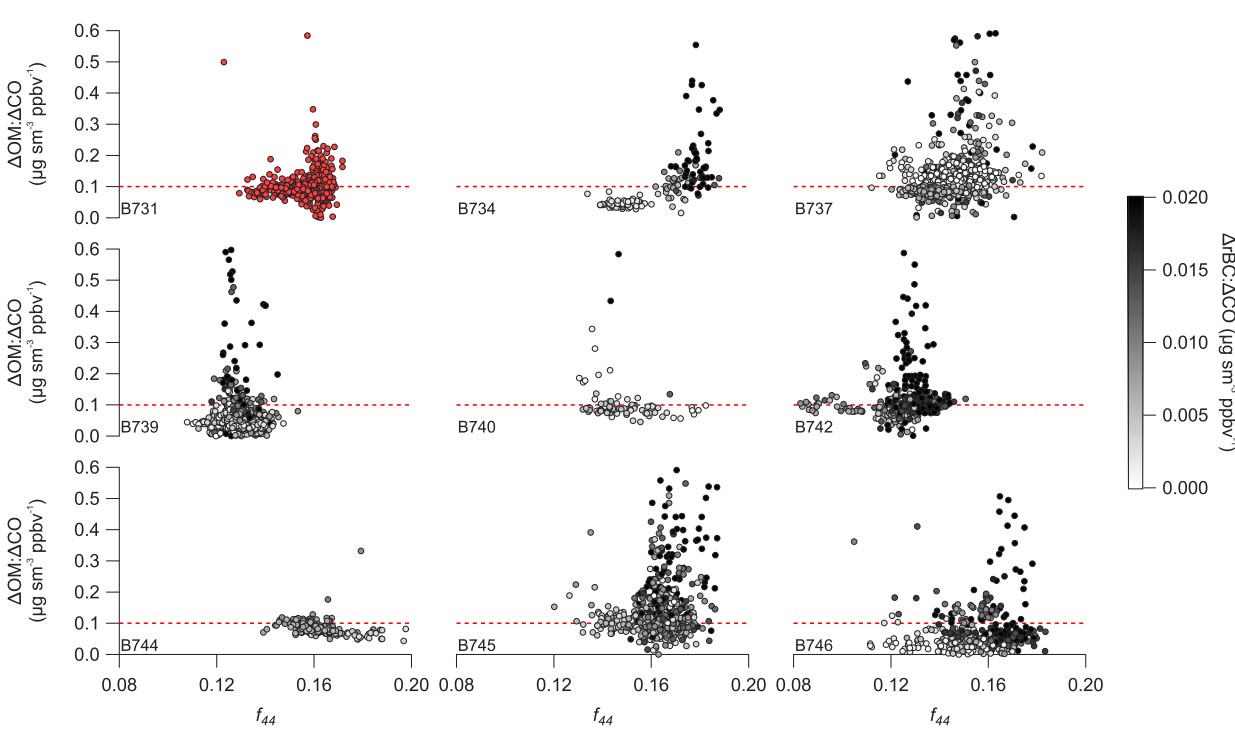

**Figure 9.** Comparison of the ratio organic matter (OM) to carbon monoxide (CO) vs $f_{44}$ across individual flights from the regional analysis. $f_{44}$ refers to the fraction of the organic aerosol mass signal at a mass-to-charge ratio of 44 measured by the aerosol mass spectrometer. Points are coloured according to the ratio of refractory black carbon (rBC) to carbon monoxide (CO) except for B731 where limited rBC data was available. The red dashed line shows the $0.1\,\mu\mathrm{g\,sm^{-3}\,ppbv^{-1}}$ as a consistent baseline for context across all flights.





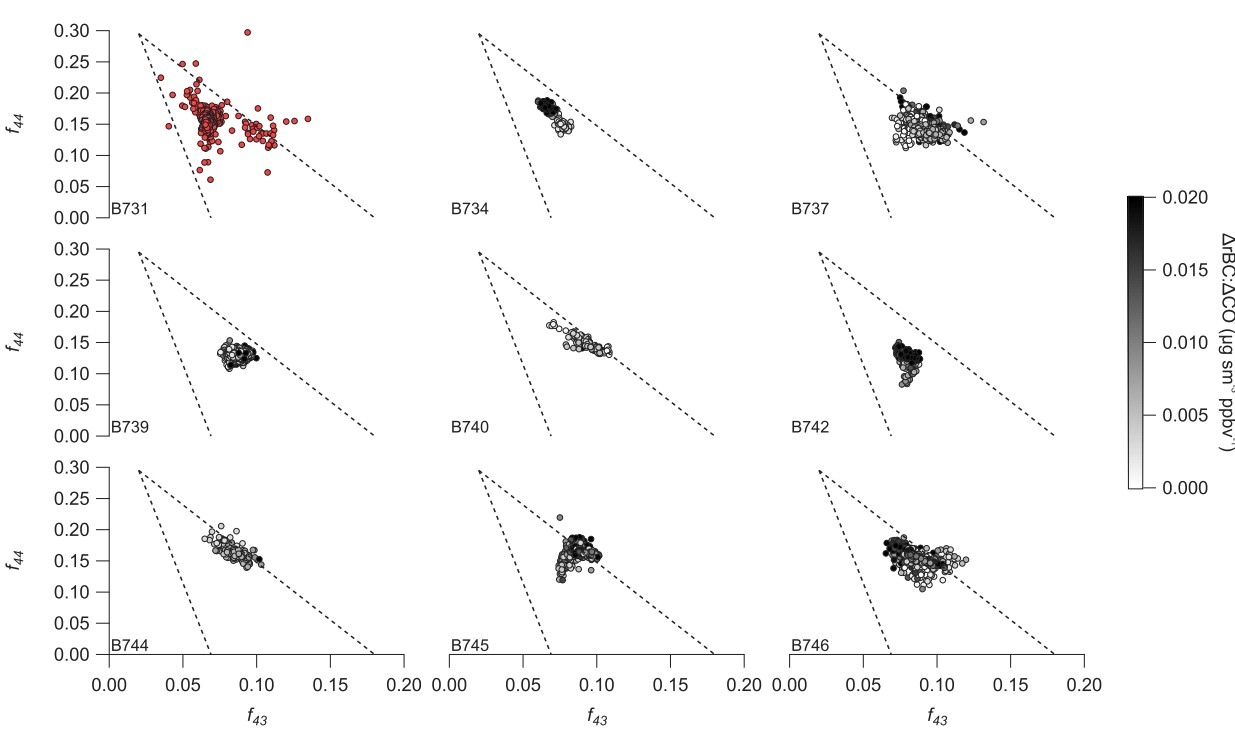

**Figure 10.** Comparison of $f_{44}$ vs $f_{43}$ from the regional analysis across individual flights, where $f_x$ refers to the fraction of the organic aerosol mass signal at a given mass-to-charge ratio measured by the aerosol mass spectrometer. Points are coloured according to the ratio of refractory black carbon (rBC) to carbon monoxide (CO) except for B731 where limited rBC data was available. Dashed lines show the 'triangle' space reported by prior studies focussed on organic aerosol less influenced by biomass burning (Morgan et al., 2010; Ng et al., 2010).





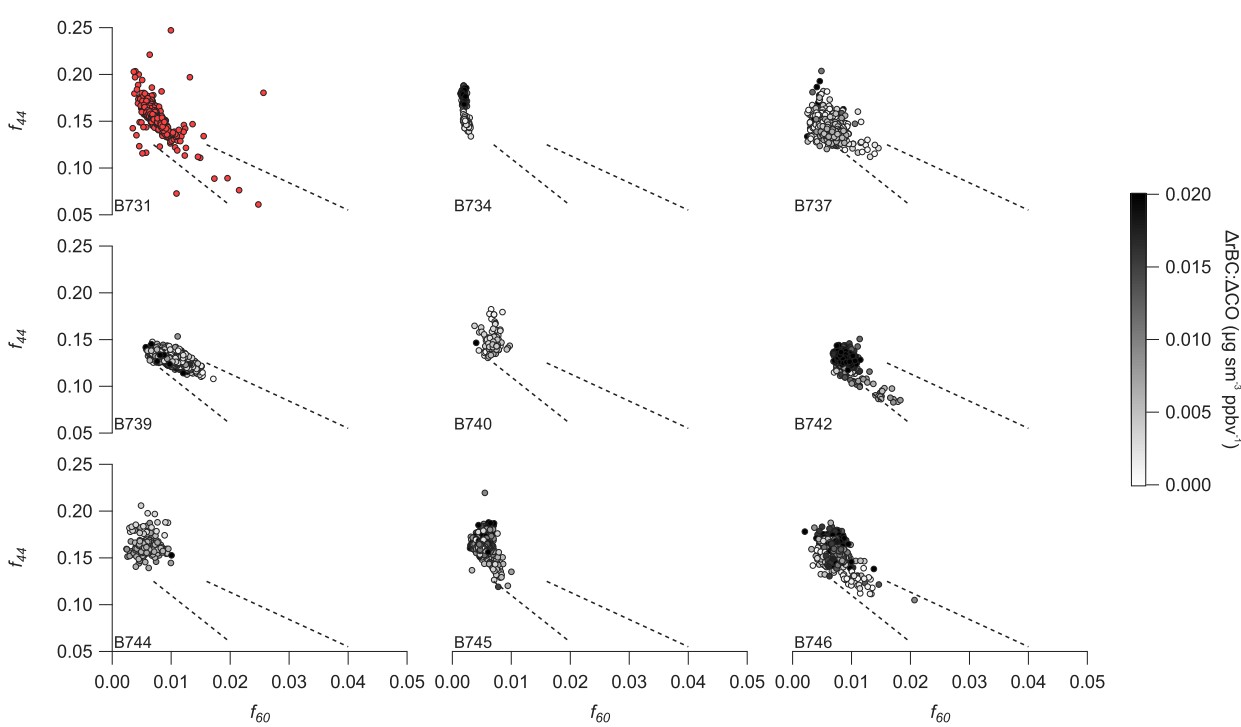

**Figure 11.** Comparison of $f_{44}$ vs $f_{60}$ from the regional analysis across individual flights, where $f_x$ refers to the fraction of the organic aerosol mass signal at a given mass-to-charge ratio measured by the aerosol mass spectrometer. Points are coloured according to the ratio of refractory black carbon (rBC) to carbon monoxide (CO) except for B731 where limited rBC data was available. Dashed lines are from previous studies on biomass burning organic aerosol aging, with the upper line from Cubison et al. (2011) and the lower line from Lack et al. (2013).



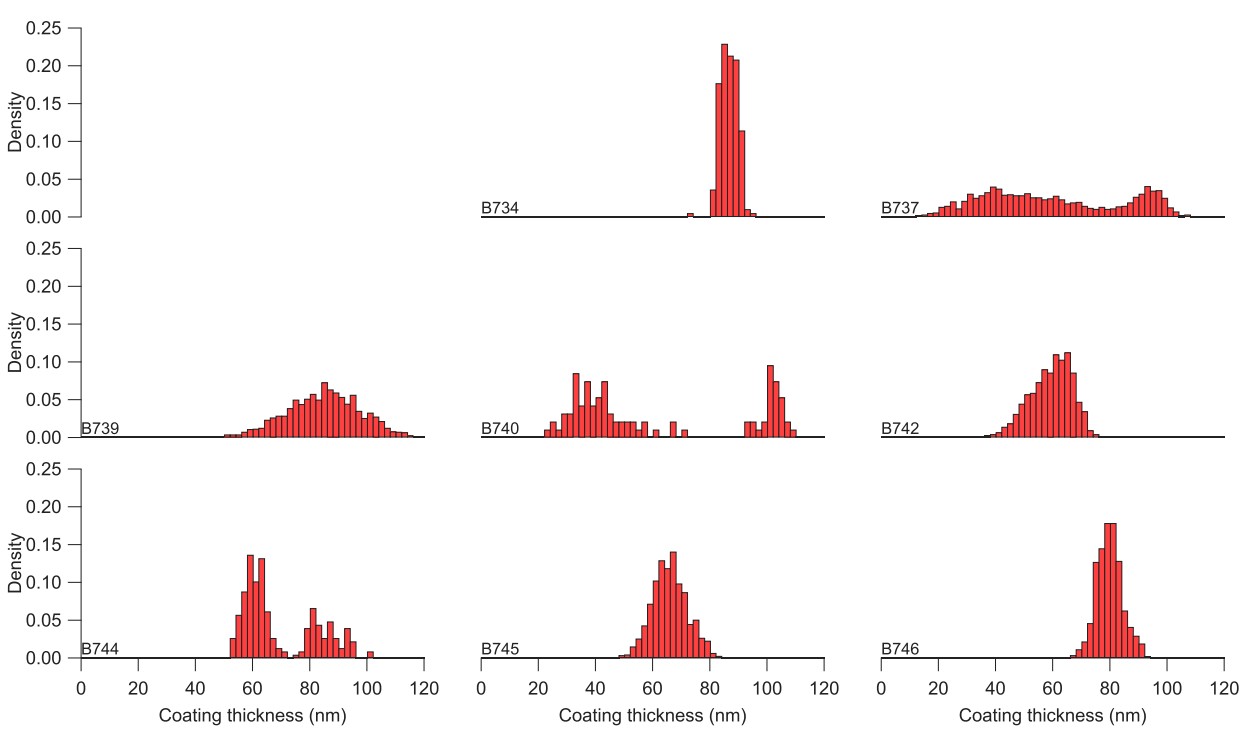

**Figure 12.** Histograms of refractory black carbon coating thickness from the regional analysis across individual flights.





**Table 1.** Flight summary of the operations included in this study. All flights were conducted during 2012. Local time is UTC-4. Take-off and land times include airport used: PVH - Porto Velho, PMW - Palmas.

| Flight | Date | Take-off (L) | Land (L) | Phase | Operating region |
|--------|------|--------------|----------|-------|------------------|
| B731 | 14 September | 10:00 (PVH) | 14:35 (PVH) | P1 | Rondônia |
| B734 | 18 September | 08:00 (PVH) | 12:15 (PVH) | P1 | Rondônia |
| B737 | 20 September | 10:45 (PVH) | 14:45 (PVH) | P1 | Rondônia |
| B739 | 23 September | 08:00 (PVH) | 12:00 (PVH) | P2 | Rondônia |
| B740 | 25 September | 07:45 (PVH) | 11:00 (PVH) | P2 | Rondônia |
| B742 | 27 September | 09:00 (PMW) | 12:30 (PMW) | P2 | Tocantins |
| B744 | 28 September | 09:00 (PVH) | 12:30 (PVH) | P2 | Rondônia |
| B745 | 28 September | 14:00 (PVH) | 17:30 (PVH) | P2 | Rondônia |
| B746 | 29 September | 09:00 (PVH) | 13:00 (PVH) | P2 | Rondônia/Mato Grosso |