# Peer review of "Transformation and aging of biomass burning carbonaceous aerosol over tropical South America from aircraft in-situ measurements during SAMBBA"

_Atmospheric Chemistry and Physics, 2019_

## Referee Comment (RC1) · Anonymous Referee #1 · 29 Mar 2019

This paper describes the analysis and interpretation of biomass burning carbonaceous aerosol emissions over Brazil from an individual plume case study and from regional haze. These emissions are thought to comprise a large fraction of the global aerosol budget and are important in their radiative effects. As these emissions age, there is a potential for changes of organic aerosol mass (OA or OM) relative to the standard inert tracer carbon monoxide (CO) due to evaporation (decrease) and secondary production (increase). The work presented here is from a unique dataset and is particularly relevant to current investigations into how biomass burning emissions are incorporated and

treated in global models. Therefore, the paper is of interest to the readership of ACP. However, the conclusions could be made more strongly if a few more details about the observations were included as described below.

One of the main conclusions is that the observed regional differences in the relationships between OM and refractory black carbon (rBC) with CO are due to the initial emissions from the fire source. However, the data actually presented in this manuscript only partially support this conclusion. The manuscript presents a very clear aging case study of OM from only a single, tropical fire with an initial, low, modified combustion efficiency (MCE) of about 0.79. The ratio of OM to CO for this fire did not change significantly from the source emissions (Figure 3) and is quite similar to the average regional haze OM for the entire study (Figure 8). The near-fire emission paper using the same dataset (Hodgson et al., 2018) included fires from the Cerrado region that had higher average MCE (about 0.94) with roughly half of the OM/CO initially emitted compared to the tropical fires (Table 5 of that paper). The current manuscript shows that the ratio of OM to CO for haze from the same Cerrado region (flight B742) is similar to that observed for the tropical fire case study (Figure 8). While this implies that a Cerrado fire ages from a lower OM to CO ratio to a higher one, this case was not presented as a contrast to the low MCE, tropical fire. It is difficult to see that any regional haze OM to CO ratio differences are directly related to differences in fire source emissions. Hence, this conclusion of the manuscript would be stronger if it included an additional aging case study of a fire with a relatively-high MCE and how the aged aerosols from such a fire transformed into regional haze.

While the analysis of the organic fraction of the biomass burning aerosols was fairly thorough, the analysis of rBC in this paper was surprisingly very limited. It was restricted mostly to coating thickness and some information on delta(rBC)/delta(CO) for the regional haze. rBC (number or mass median?) core sizes were mentioned in the abstract and some of the text, yet no data were presented. Furthermore, the sizes reported here were very different (250-290 nm) from those presented in the Hodgson

work (number medians around 100 nm and mass medians around 200 nm). It looks like the mass of rBC in the fire overpass intercepts (and associated background) were significantly higher than in the along plume straight-and-level run (SLR) (Figure 2). There did not seem to be any explanation for this or how it might impact the derivation of a "background" rBC mass for the delta(rBC) calculations. More rBC data (not just in the text but in figures) and analysis should be included in this manuscript (e.g., mass of rBC, number fraction of rBC particles to the total number of particles, rBC-core size, and number fractions of rBC particles with coatings compared to all rBC particles as a function of plume age and similar values for the regional haze). Also, there was no data showing changes in the ratio of rBC to CO with wet scavenging. Such additional information about the rBC-containing carbonaceous aerosols should be included in this manuscript.

Minor comments:

P1 L6 and elsewhere: OA was used here and OM was used in other places. Consider using one or the other.

P1 L17-19: If supported with additional analysis, this last sentence is confusing and should be re-written more clearly.

P4 L17: change "refractory" to "non-refractory"

P5 L32: should "cross-plume" be in front of "SLR" here?

P5 L44 and elsewhere: The numbers for several figures are missing in the text.

P5 L51: Add citation to the Hodgson paper for the 97.1% value.

P6 L25: How does the time to reach background levels (3 hours) compare with calculated entrainment/mixing rates base on the boundary layer heights and the atmospheric (in)stability?

P6 L38-41: What were the delta(rBC)/delta(CO) ratios and rBC core diameters referred

to here? Suggest showing in SI.

P7 L48 and others shortly afterward: "absolute difference and correlation between the mass spectrum and the background" – does this refer to deviation of the slope from one and r-value from a linear least squares correlation obtained by plotting the absolute peak intensities (instead of the "fraction of organic signal" peak intensities) of the top spectrum versus the bottom spectrum in Figure 4? Please clarify.

P8 L44 and after: Were there any flights where there was a clear distinction in delta(rBC)/delta(CO) before and after raining? This would be interesting to explore in light of the assertions made here about air mass history and fire burning conditions.

P9 L15 and earlier: The fraction of rBC particles that were coated is discussed in this paper, but the actual fraction is not shown in this paper. Should be included some-where.

P9 L22-23: Should include the rBC core-size data somewhere.

P10 L43 & 44: "limited net enhancement of OA" seems to be inconsistent with "net OA production". Please clarify.

P10 L54-56: This last sentence was confusing; suggest revising. Perhaps add a sec-ond statement (if observed) about how the rBC mass was useful for distinguishing differences in air mass history for two similar instances of OA/CO with the same f44.

P11 L34-35: This statement is made without mentioning the actual fractions of coated rBC particles in the different locations studied.

P11 L54: consider adding to the end of this sentence "to essentially background val-ues."

Figure 1: It was unclear how the flight track matched up with the map. Suggest enlarg-ing the map to show where the fire was.

Figure 2: The values may be easier to obtain/visualize directly from the plots if they had

horizontal lines across them. It would be good to see where the background CO, OM, and rBC values were chosen, especially for the along-plume SLR. Consider having plots of OM vs CO, rBC vs CO, and Inorganics vs CO in the SI. Add the fire altitude to the caption. Consider expanding the altitude axis and including the ground level height.

Figure 3: This could be split into three separate figures: one of the top three plots, one of f44, f43, and f60 vs. plume age (instead of Figure 5), and one of the rBC data (delta(rBC)/delta(CO), rBC median number core size, fraction of rBC particles coated, and rBC coating thickness) vs. plume age. All three of these should have "background" values on the right hand side plus both horizontal and vertical grid lines for reference. Consider adding data from the cross-plume intersect points too.

Figure 4: The background mass loading from "regional aerosol haze away from the main fire plume study region" is 4.1 here, but in Figure 7 the average background mass loading for this flight was about 10. Why were they so different?

Figure 5: It is not clear why the dashed lines are important. How would these plots look if only mixing was occurring? Suggest omitting this figure or relegating to SI.

Figure 6: Suggest enlarging the map showing only the portions of the flight tracks where the regional haze data were obtained and labeling the states with regional haze flight data. For the enlarged map it would be helpful to have an indication of scale.

Figure 7: Consider adding the location information from Table 1 to the caption, or at least mention which two flights were not only over Rondonia. Maybe point out the flight numbers on the map in Figure 6?

Figure 8: It is difficult to distinguish variation in delta(rBC)/delta(CO) because the points are outlined in black. Suggest including the average value for each delta(OM)/delta(CO) value shown. Does B737 show the opposite trend (higher delta(OM)/delta(CO) with lower delta(rBC)/delta(CO)) than flights B734 and B746 (with lower delta(OM)/delta(CO) for lower delta(rBC)/delta(CO))? It's difficult to see either

trend for flight B739 – are both those slopes for relatively low delta(rBC)/delta(CO)? Is there another indicator that could be used to distinguish low delta(rBC)/delta(CO) arising from precipitation rather than burn conditions? Perhaps the B737 flight was more/less impacted by precipitation than the other flights? Also, consider a table in the SI of background OM and CO (and rBC) for each correlation shown.

Figure 9: How do low delta(CO) values contribute to the uncertainty in the high delta(OM)/delta(CO) ratios?

Figure 12: Suggest adding tick marks on the x-axis for the flights in the top and middle rows. Also consider coloring each bar with the average delta(rBC)/delta(CO) ratios for that coating thickness bin.

Table 1: (L) must mean Airport. What does Phase P1 or P2 mean?

References

Hodgson, A. K., Morgan, W. T., O'Shea, S., Bauguitte, S., Allan, J. D., Darbyshire, E., Flynn, M. J., Liu, D., Lee, J., Johnson, B., Haywood, J. M., Longo, K. M., Artaxo, P. E., and Coe, H.: Near-field emission profiling of tropical forest and Cerrado fires in Brazil during SAMBBA 2012, Atmos. Chem. Phys., 18, 5619-5638, https://doi.org/10.5194/acp-18-5619-2018, 2018.

---

## Referee Comment (RC2) · Anonymous Referee #2 · 23 Apr 2019

The present manuscript is a highly-focused piece of work that examines chemical and microphysical property changes of biomass burning aerosol emissions that were sampled during the SAMBBA field campaign in Brazil. By and large the manuscript is written reasonably clear (explicit areas where more clarity is needed are highlighted below). As biomass burning (BB) events represent a major source of particulate matter injected into the atmosphere, it is indeed important to characterize the emissions and to understand how the chemical, microphysical, and optical properties transform as the plume ages so that models can, in turn, improve upon there fidelity. Using the

analysis tool of excess mixing ratios, the authors report on a negligible net increases in organic aerosol (OA) even though there is concomitant changes in chemical properties - namely the oxidation state of the OA. This finding is inline with that reported by others and the authors put forth the same argument as that put forth by others that the loss of primary organic aerosol (POA) due to plume dilution is offset by the production of secondary organic aerosol (SOA), hence the negligible change in net OA loading. The other reported findings centers on refractory black carbon and its mixing state. In summary, the present manuscript adds important findings to a growing body of data on biomass burn-generated carbonaceous aerosol production and evolution and as such should be published. That said, to this reviewer at least, this manuscript has a feel of being incomplete in some of its analyses and in other areas left this reviewer wondering why other datasets were not part of this manuscript. This feeling could be simply that the authors are parsing out various subject matter for other manuscripts (e.g., optical properties). If this is indeed the case, the authors should be explicit about that. Once this authors address this ad some other top-level items listed below along with several specific items (e.g., figure numbering and clarification text) the manuscript should be acceptable for publication.

While this manuscript is highly-focused, as highlighted above, it seems that the authors miss the opportunity to role in other data sets that could, potentially, elucidate all that is going on in these complex emissions plumes. For example, the authors report that there are changes in the OA chemical properties and that they observe that the rBC coating thickness changes as a function of plume age, yet authors do not report on, nor reference any parallel paper, that looks at aerosol size distribution - arguably one of the most fundamental of measurements in our business. Doing so could tell us something about the roll of coagulation near the fire source as well as informing us about how the distribution evolves. Does the mode stay constant or grow?As hinted at above, perhaps the authors have the intention of publishing a separate aerosol size distribution-centric paper. If this is indeed the case they should make reference it, as the analysis of these dataset would easily complement what is being learned from

the AMS. The absence of this dataset is even more puzzling given that the authors report on geometric mass mean diameter for rBC - along with estimates of the coating thicknesses. Why present microphysical properties of only one species (rBC) and not the primary particulate species (OA)?

Similarly, the authors say nothing about aerosol light scattering or light absorption. How does the scattering Angstrom exponent evolve with plume age in the near field? Does the mass scattering efficiency track what is observed with the AMS? In their rBC coating thickness analysis, the authors assume that the coating this transparent (coating refractive index of 1.5 +0i). What is the basis for this assumption? BB events are a known source of brown carbon. What does the absorption Angstrom exponent suggest? And, of course, what is the SSA doing in these first few hours where a lot of chemistry appears to be going on? As with the size distribution, perhaps the authors will report on this in a separate manuscript. It just seems to this reviewer that the absence of any reference to either the optical or size distribution datasets misses any opportunity to better examine what is going on.

In their examination of regional BB haze, the authors use the AMS tracer f44 and the ratio of rBC to CO. As the author state, rBC and CO are relative inert tracers that are strongly controlled by initial conditions, and that the ratio can provide some information about the influence of precipitation. But my question to the authors, especially when examining regional haze, is how do you know what the initial ratio was at the various sources that are contributing to the haze? Are all fires assumed to exhibit the same burning phase conditions (e.g., flaming versus smoldering)? Figure 7 indicates that the mass fraction of rBC ranged from 2% to 12.4%. Are the authors saying that the burn conditions are the same for these two bounding conditions? Here is where I would have expected some discussion of modified combustion efficiency (MCE) which might help answer this by telling us something about the initial burn conditions. Under active flaming conditions little CO is produced and more rBC while under smoldering conditions, more CO is produced and little rBC. Could this explain the variability observed in

rBC mass fraction contributions or is the variability driven by differences in source fuel or subsequent cloud processing (e.g., rBC loss through precipitation)? While not listed in this paper, Darbyshire reports that a CO2 analyzer was deployed on the FAAM and thus is presumably available to use along with the CO data set, to estimate the MCE. It might b interesting to see if any of the variability in rBC mass fraction contributions can be explained by MCE. And if so, maybe the MCE can, in turn, improve the robustness of the rBC/CO ratio analysis. Finally, the authors might want to check out Collier et al., (figure 4; EST, 50, 8613, 2016) who reported on the relationship between OA production and MCE - where OA production was favored under smoldering conditions.

The authors state that the fire used in their case study was likely a natural fire and one that is "highly-smoldering". Do we expect a highly-smoldering fire to generate a 12.4% mass fraction of rBC?

The manuscript seems heavy on the figures (12 figures). The authors are encouraged to try to reduce this number.

Specific comments

Abstract: page 1, Line 32: Please add "mean mass diameter" in front of "250- 290 nm".

Page 4. Line 11: Please add figure number. ("1")

Page 4, Line 22: Please add figure number ("1"). Also, not clear what comes after "and"

Page 4, Line 30: Please add figure number ("2")

Page 4, Line 35: Please add figure number ("3")

Page 5, Line 4-5: How do the authors explain a slight decrease in rBC core diameter? This is a curious finding and this reviewer cannot help but wonder if the reported decrease is due to a measurement artifact as this was reported for the "case" study where the plume lifecycle could be well constrained (i.e., no cloud processing of coated

rBC particles that could selectively wash out larger diameter rBC-containing particles). What were the highest number concentrations encountered near the source? Back-of-the-envelope calculations assuming a size mode of 125 nm rBC particle and 1.5 ug/m3 suggests < 1000 particles/cc which should be low enough not to suffer from particle coincidence. Again, this is a very curious finding to simply close out a paragraph with, with no follow on statement or discussion.

Page 5, Line 7: Please add figure number ("4")

Page 5, Line 23: Please add figure number ("5") - Are both figures 4 and 5 necessary?

Page 5, Line 33: What is the useful "aging" range of the f44 marker as a tracer of age?

Page 6, Line 23: Please add Figure number ("5")

Page 6, Lines 34-39 and page 21, Figure 8. As discussed above, perhaps examining the MCE might provide some useful insights. The delta rBC/delta CO scale in Figure 8 nominally ranges from $\sim$0.0025 to $\sim$ 0.02. Is this variability driven by cloud processing (i.e., precipitation) or MCE.

Page 25, Figure 12: Again, not to harp on the MCE theme, but it might prove interesting to examine whether the variability in coating thickness is driven by processing or MCE.

---

## Referee Comment (RC3) · Anonymous Referee #3 · 30 Apr 2019

This paper provides an analysis of BC, OA, CO and OA oxidation state for a 2012 airborne field campaign of biomass burning emissions at Porto Velho Brazil. Data is presented in two parts, a case study of a single smoldering tropical plume and a regional analysis of 9 other regional flights. Some aspects of the paper were nicely put together, such as the observation of the evolution of smoke oxidation state. But the overall purpose of the paper on the evolution of aerosol mass is short on many important details. Trying to sort out mass evolution is quite tricky, especially for an individual plume. Accounting for the temporal evolution of the fire, controlling for combustion ef-

ficiency, linking source plumes to the regional haze, all take a great deal of care. One also needs to demonstrate appropriate cross correlation between numerous parameter to ensure an apples to apples comparison to anything about temporal evolution. This is especially true in the present paper where it is clear from the regional survey work that the smoke particle properties show a lot of heterogeneity. While, I think the authors have spent a great teal of time on this paper, I found the narrative unconvincing. I think the paper probably needs significant revisions and resubmitted. At this point I think I can keep my comments to three main themes.

1) The paper references a great deal of "Recent activity" but the whole line of scientific thought on particle evolution came out of the ABLE, ESPRESSO, SCAR-C, SCAR-B missions of the late 80's and 1990s. These studies were much more rigorous than anything that is presented here. Liousse demonstrated he issues with particle evaporation, and Martins, Reid and Hobbs evaluated secondary production and found in well documented Lagrangian plumes samples significant production. We know the authors of this paper are aware of this work because some of the co authors were actually on these papers. A summary of this work is in the 2005 biomass burning review papers by Reid. The conclusion "secondary production is complicated and varies by fire" is indeed true, but there has been a great deal of work done in the past and even currently (all un referenced) that actually narrows down processes. Reid and Martins points are that secondary production and basic condensation happen very rapidly. Secondary production of sulfate requires cloud processing. Going back to the late 1980s significant and rapid organic acid production has been observed (I think ABLE mission). This paper lacks any concrete linkage to past knowledge to move the field forward.

2) The single case study presented is for a low combustion efficiency plume without any presented evidence that downwind samples are of the same fire characteristics. During the observation of fires in SCAR-C and SCAR B it was found that fire properties change rapidly. Given that the test case had a MCE< 0.8, then black carbon production must have been at a minimum. Perhaps there was some flaming combustion along

the periphery. Therefor the relationship of secondary production to rBC is probably pretty tenuous. At the same time, most of the cases observed fo secondary production have been associated with flaming combustion. Smoldering combustion is essentially a surface reaction. So with the limited data provided, I am not sure what to make of this particular test case.

3) Both comments one and two then project onto overall issue of making an apples to apples comparison to evaluate particle evolution. The authors report an MCE value, so there must be $CO_2$ data available. But no time series of MCE is provided, nor even a CO to rBC ratio plot. Rather the reader has to do an eyeball comparison of the two on a log plot. For the regional samples we are presented with a great deal of variability in particle properties (other than well known oxidation with time) but not the types of additional data that other studies have used to sort out what is going on. I think the authors need to spend more time on the data narrative.

---

## Author Comment (AC1) · 2 Jan 2020

**Reviewer Comments**

We thank the reviewers for their comments on our manuscript, which we have sought to answer and/or amend the revised submission. In the following we have included the reviewers' comments in bold text, with our responses beneath.

Given that each of the referees has questioned the reasoning for an apparent lack of additional measurements, namely total particle number concentrations, particle size distributions and optical properties, we provide a summary of our reasoning below.

Unfortunately these measurements either suffered from complete instrument failures on several flights, or did not pass quality assurance checks when analysed post-campaign. Instrument failures were particularly acute during our case study flight, so could not be included as an extended analysis for that flight alone.

In addition to the above instrument issues, in response to reviewer #2 regarding the absence of brown carbon measurements and the absorption Angstrom exponent, our PSAP measurements were single wavelength only.

Given our focus on the wide ranging conditions both within and between flights, we decided to base our study on the AMS and SP2 measurements given the data coverage was much greater and we had confidence in the robustness of the data. The alternative would have been to use a severely compromised dataset that would have made direct comparisons between flights impossible, while significantly lengthening the paper. Furthermore, the combination of the AMS and SP2 is the major novelty of our study, and provides valuable additions to the literature even with the absence of other measurements (most of which have been more extensively studied in the existing literature).

Modified combustion efficiency (MCE) estimates were also mentioned by the referees – based on our analysis and reading of the literature, we do not find that such estimates are robust for regional studies such as ours where the enhancements of $CO_2$ above the background level are often minimal and/or impossible to quantitatively estimate. In fact, we did not have sufficient confidence to use such estimates for the plume-run portion of our study as the uncertainties were large and rendered the MCE values as effectively non-quantitative. Similarly to the discussion regarding inclusion of aerosol microphysical and optical measurements, we instead have focussed on the novel measurements where we are confident in their robustness. Given this, we use MCE for near-field situations only, where we have greater confidence in both the CO and $CO_2$ measurements and determination of their background values.

In summary, the measurements presented in our study are what we have available and provide a valuable and unique addition to the literature, even without the additional measurements suggested by the reviewers. We cannot change what instruments and data are available – there are significant challenges to conducting airborne campaigns of this size in a remote environment where heat and humidity present sizeable obstacles for instrument operations.

**Anonymous Referee #1**

This paper describes the analysis and interpretation of biomass burning carbonaceous aerosol emissions over Brazil from an individual plume case study and from regional haze. These emissions are thought to comprise a large fraction of the global aerosol budget and are important in their radiative effects. As these emissions age, there is a potential for changes of organic aerosol mass (OA or OM) relative to the standard inert tracer carbon monoxide (CO) due to evaporation (decrease) and secondary production (increase). The work presented here is from a unique dataset and is particularly relevant to current investigations into how biomass burning emissions are incorporated and treated in global models. Therefore, the paper is of interest to the readership of ACP. However, the conclusions could be made more strongly if a few more details about the observations were included as described below.

One of the main conclusions is that the observed regional differences in the relationships between OM and refractory black carbon (rBC) with CO are due to the initial emissions from the fire source. However, the data actually presented in this manuscript only partially support this conclusion. The manuscript presents a very clear aging case study of OM from only a single, tropical fire with an initial, low, modified combustion efficiency (MCE) of about 0.79. The ratio of OM to CO for this fire did not change significantly from the source emissions (Figure 3) and is quite similar to the average regional haze OM for the entire study (Figure 8). The near-fire emission paper using the same dataset (Hodgson et al., 2018) included fires from the Cerrado region that had higher average MCE (about 0.94) with roughly half of the OM/CO initially emitted compared to the tropical fires (Table 5 of that paper). The current manuscript shows that the ratio of OM to CO for haze from the same Cerrado region (flight B742) is similar to that observed for the tropical fire case study (Figure 8). While this implies that a Cerrado fire ages from a lower OM to CO ratio to a higher one, this case was not presented as a contrast to the low MCE, tropical fire. It is difficult to see that any regional haze OM to CO ratio differences are directly related to differences in fire source emissions. Hence, this conclusion of the manuscript would be stronger if it included an additional aging case study of a fire with a relatively-high MCE and how the aged aerosols from such a fire transformed into regional haze.

While we agree with the reviewer that the manuscript would be stronger if it included an additional aging case study of a fire with higher MCE, we simply do not have such measurements. The fires in the Cerrado were typically much smaller in size and the amount of smoke quickly dissipated to sub-visible plumes; we were able to sample the above-fire plumes in the near-field, which was the focus of Hodgson et al., 2018, but plume aging measurements examining the transition from the near-field to regional haze were impossible.

We would also add that directly comparing the regional and the near-field measurements over the Cerrado is complicated by the regional aerosol likely being from a range of sources that have been advected and mixed. This is particularly relevant in the Para/Mato Grosso/Tocantins borders as the vegetation and fire characteristics have larger gradients than in the west. As such we do not focus on such comparisons as attempting to separate

the sources e.g. via dispersion modelling, is beyond the scope of our measurement-focussed study.

**While the analysis of the organic fraction of the biomass burning aerosols was fairly thorough, the analysis of rBC in this paper was surprisingly very limited. It was restricted mostly to coating thickness and some information on delta(rBC)/delta(CO) for the regional haze. rBC (number or mass median?) core sizes were mentioned in the abstract and some of the text, yet no data were presented. Furthermore, the sizes reported here were very different (250-290 nm) from those presented in the Hodgson work (number medians around 100 nm and mass medians around 200 nm). It looks like the mass of rBC in the fire overpass intercepts (and associated background) were significantly higher than in the along plume straight-and-level run (SLR) (Figure 2). There did not seem to be any explanation for this or how it might impact the derivation of a "background" rBC mass for the delta(rBC) calculations. More rBC data (not just in the text but in figures) and analysis should be included in this manuscript (e.g., mass of rBC, number fraction of rBC particles to the total number of particles, rBC-core size, and number fractions of rBC particles with coatings compared to all rBC particles as a function of plume age and similar values for the regional haze). Also, there was no data showing changes in the ratio of rBC to CO with wet scavenging. Such additional information about the rBC-containing carbonaceous aerosols should be included in this manuscript.**

We contest the reviewers' assertion that our rBC analysis was 'very limited' – rBC mass, coating thickness and rBC:CO ratios feature in every figure of the manuscript aside from the maps and the two AMS mass spectra-related figures from the plume case study. The ratio of rBC:CO in particular is a crucial aspect of our analysis and we provide far greater detail showing the range of values we observe, rather than bulk averages as has been common in the existing literature.

We did not include as much rBC size distribution data as the number of figures in the manuscript was already large (as noted by reviewer #2 who suggested reducing the number of figures) and we felt that we could sufficiently describe the results in the text body of the paper.

Several of the derived measurements suggested by the reviewer relating to comparing rBC number to total particles are not feasible from our dataset due to the instrument issues mentioned in our introductory response. We do not have direct measurements of wet scavenging and can only present our observed rBC:CO ratios and coating thickness across each flight – as we note in the manuscript, we use the changes in rBC:CO ratio to infer the presence of wet scavenging of the regional aerosol; without a different direct measurement, examining the rBC:CO ratio as suggested by the reviewer would be a circular endeavour.

In response to comment re. P6 L38-41 below, we have added a direct comparison of the rBC core sizes from both the above-fire intercepts and plume-run finding mean sizes of 270 nm and 249 nm respectively, with significant variability in each. In this paper we have reported the geometric mean diameter, whereas Hodgson et al. reported the diameters from log-normal fits on the average distributions from the in-plume sampling, so the reviewer is not drawing an apples-to-apples comparison.

The reviewer notes that the rBC mass in the fire overpass intercepts were significantly higher than the along-plume sampling – given the distance and time since emission between the two, dilution and mixing with cleaner air away from the immediate vicinity of the fire are likely key in driving the observed peak and background mass values.

In summary, we feel that we have presented a great deal of detail on rBC-containing carbonaceous aerosol that is new to the literature both in this region and more generally. Had we had the additional measurements required to present the further suggestions from the reviewer, then we would have included them.

**Minor comments:**
**P1 L6 and elsewhere: OA was used here and OM was used in other places. Consider using one or the other.**

We have changed the majority of uses of OA to OM throughout the manuscript. We did retain OA in some cases where it was more appropriate e.g. for other studies that didn't specifically measure OM. We also retained the use of OA in the abstract as a more generic "catch-all" term.

**P1 L17-19: If supported with additional analysis, this last sentence is confusing and should be re-written more clearly.**

Rephrased.

**P4 L17: change "refractory" to "non-refractory"**

Amended.

**P5 L32: should "cross-plume" be in front of "SLR" here?**

Amended.

**P5 L44 and elsewhere: The numbers for several figures are missing in the text.**

This was a result of incorrectly referencing the Figures in our LaTeX file – we have amended the references to fix the issue.

**P5 L51: Add citation to the Hodgson paper for the 97.1% value.**

Added.

**P6 L25: How does the time to reach background levels (3 hours) compare with calculated entrainment/mixing rates base on the boundary layer heights and the atmospheric (in)stability?**

We have investigated whether we can reliably use the aircraft meteorological data to estimate the entrainment rate. However, due to the flight path followed and the limitations on aircraft duration the necessary information was not available.

**P6 L38-41: What were the delta(rBC)/delta(CO) ratios and rBC core diameters referred to here? Suggest showing in SI.**

We have added the values to the text, while noting that the somewhat lower values in the plume-run compared to the above-fire intercepts possibly point to a slight shift in fire conditions, although the variability is large in both cases making it likely unwise to draw definitive conclusions.

**P7 L48 and others shortly afterward: "absolute difference and correlation between the mass spectrum and the background" – does this refer to deviation of the slope from one and r-value from a linear least squares correlation obtained by plotting the absolute peak intensities (instead of the "fraction of organic signal" peak intensities) of the top spectrum versus the bottom spectrum in Figure 4? Please clarify.**

We have added a note to the paragraph to clarify what is being reported.

**P8 L44 and after: Were there any flights where there was a clear distinction in delta(rBC)/delta(CO) before and after raining? This would be interesting to explore in light of the assertions made here about air mass history and fire burning conditions.**

We agree that such an examination would be of interest but we do not unfortunately have observations to address this effect – our flight times were relatively short due to the operational restrictions that are unavoidable when working in such a remote location, as well as factoring in transit times that further cut down on low-level flying.

**P9 L15 and earlier: The fraction of rBC particles that were coated is discussed in this paper, but the actual fraction is not shown in this paper. Should be included somewhere.**

There appears to be some confusion on the nature of our measurements – we present coating thicknesses for rBC particles, not the fraction of rBC particles that were coated. The clearest example of this is Fig. 12 where we show the density of rBC particles with a given thickness – as discussed in the paper, we typically see coatings of greater than 40nm, with the lowest coatings being in the 10-20nm range during B737. Our measurements imply that the rBC is at the very least partially coated across our dataset i.e. the "fraction" is close to unity but the thickness of such coatings does vary.

**P9 L22-23: Should include the rBC core-size data somewhere.**

We did not include figures illustrating the data as there was little value in them due to the limited variability we note in the text, thus such figures would be fairly redundant and add little that could not be communicated in the main text.

**P10 L43 & 44: "limited net enhancement of OA" seems to be inconsistent with "net OA production". Please clarify.**

Amended.

**P10 L54-56: This last sentence was confusing; suggest revising. Perhaps add a second statement (if observed) about how the rBC mass was useful for distinguishing differences in air mass history for two similar instances of OA/CO with the same f44.**

We have added that rBC:CO was the key diagnostic for distinguishing differences in air mass history to make our statement clearer.

**P11 L34-35: This statement is made without mentioning the actual fractions of coated rBC particles in the different locations studied.**

We have added some additional commentary to the text detailing the observed thickness of the particles, that indicates that the majority of rBC particles are at least partially coated with the lowest observed thicknesses in the 10-20nm range. We also refer to the response above regarding P9 L15.

**P11 L54: consider adding to the end of this sentence "to essentially background values."**

Based on Figs. 4 and 5, we have amended the suggested statement as the data implies there is potentially scope for further aging to "essentially background levels" – in the prior analysis and discussion in the main text we referred to the values being "comparable/resembling" without going as far as to say they had reached background levels, which is not categorically supported by the observations.

We have added "…and a comparable magnitude to the background regional aerosol" to the main text.

**Figure 1: It was unclear how the flight track matched up with the map. Suggest enlarging the map to show where the fire was.**

We have enlarged the map to better illustrate the flight's location.

**Figure 2: The values may be easier to obtain/visualize directly from the plots if they had horizontal lines across them. It would be good to see where the background CO, OM, and rBC values were chosen, especially for the along-plume SLR. Consider having plots of OM vs CO, rBC vs CO, and Inorganics vs CO in the SI. Add the fire altitude to the caption. Consider expanding the altitude axis and including the ground level height.**

We have added horizontal lines to the plots to aid reading the figure. Adding the sections where the background values were determined would add significant clutter to the figure, so we have omitted these details. Regarding the along-plume SLR specifically, a short SLR in clear air from 1225-1230 was used prior to the aircraft repositioning to conduct the plume run.

**Figure 3: This could be split into three separate figures: one of the top three plots, one of f44, f43, and f60 vs. plume age (instead of Figure 5), and one of the rBC data (delta(rBC)/delta(CO), rBC median number core size, fraction of rBC particles coated, and rBC coating thickness) vs. plume age. All three of these should have "background" values on the right hand side plus both horizontal and vertical grid lines for reference. Consider adding data from the cross-plume intersect points too.**

We have added horizontal lines to the plots to aid reading the figure.

Adding cross-plume intersect points and "background" values would complicate the figure more than is really necessary – the aim is to show the plume-run data and compare it to the above-fire intercepts, not show every data-point. We did previously experiment with adding the background values but this was largely superfluous as the data beyond 70 km were essentially showing the same information.

As discussed below, we have not removed Figure 5 and the suggestions by the reviewer would swell the number of figures unnecessarily as we have captured the details of the additional parameters (where measured) in the text. Additionally, reviewer #2 noted that the paper is "heavy on the figures" – we agree and this motivated our decision to not include more of them.

N.B. While reviewing Fig. 3, we realised that we had made an error when transferring the Hodgson et al. OM, nitrate and sulfate ratios – we had failed to convert to the units we use in the figure, whereas in Hodgson et al., the molar ratios are reported. The corrected values are a factor of 1.23 larger as a result in the updated figure. We have also added that the values are from Hodgson et al. in the figure caption.

**Figure 4: The background mass loading from "regional aerosol haze away from the main fire plume study region" is 4.1 here, but in Figure 7 the average background mass loading for this flight was about 10. Why were they so different?**

Fig. 7 doesn't show the "average background mass loading" as stated by the reviewer – it shows the average regional loading across the flight which is subject to e.g. regional scale advection of the aerosol burden. The background mass loading referred to in Fig. 4 is just one portion of the study in the vicinity of the fire we studied, which was evidently a somewhat cleaner environment than the region to the north where the rest of the regional measurements were primarily conducted.

**Figure 5: It is not clear why the dashed lines are important. How would these plots look if only mixing was occurring? Suggest omitting this figure or relegating to SI.**

The dashed lines are from prior studies and are useful for understanding how our measurements relate to such work and are referred to in the text. Their addition is to illustrate that our measurements fall within the range of prior AMS observations, which is a useful comparison. Furthermore, having presented this exact figure at conferences attended

by others in the AMS community, we had valuable discussions as a result and strongly feel the figure is an important part of our work that belongs in the main manuscript.

**Figure 6: Suggest enlarging the map showing only the portions of the flight tracks where the regional haze data were obtained and labeling the states with regional haze flight data. For the enlarged map it would be helpful to have an indication of scale.**

We have enlarged the map and included an inset figure showing the whole of South America to illustrate the flight locations. We have also added a scale for the enlarged map and text labels for the different states.

**Figure 7: Consider adding the location information from Table 1 to the caption, or at least mention which two flights were not only over Rondonia. Maybe point out the flight numbers on the map in Figure 6?**

We have added the flight locations to the caption and added labels for B742 & B746 to Fig. 6.

**Figure 8: It is difficult to distinguish variation in delta(rBC)/delta(CO) because the points are outlined in black. Suggest including the average value for each delta(OM)/delta(CO) value shown. Does B737 show the opposite trend (higher delta(OM)/delta(CO) with lower delta(rBC)/delta(CO)) than flights B734 and B746 (with lower delta(OM)/delta(CO) for lower delta(rBC)/delta(CO))? It's difficult to see either trend for flight B739 – are both those slopes for relatively low delta(rBC)/delta(CO)? Is there another indicator that could be used to distinguish low delta(rBC)/delta(CO) arising from precipitation rather than burn conditions? Perhaps the B737 flight was more/less impacted by precipitation than the other flights? Also, consider a table in the SI of background OM and CO (and rBC) for each correlation shown.**

We have altered the colour scheme on the plot to make comparisons easier. The red text shows the slope of the line-of-best-fit for dOM vs dCO, which is essentially the average value, which was suggested for inclusion by the reviewer.

As discussed elsewhere, we do not have a clear and quantitative means of separating the influence of precipitation from burn conditions.

N.B. We have updated the colour scheme on Figs. 9-11 to match that used for Fig. 8.

**Figure 9: How do low delta(CO) values contribute to the uncertainty in the high delta(OM)/delta(CO) ratios?**

The reviewer raises a good point here – at least some of the extreme dOM/dCO values is likely driven by low concentrations of OM and/or CO. We have added a comment to the text to highlight this.

**Figure 12: Suggest adding tick marks on the x-axis for the flights in the top and middle rows. Also consider coloring each bar with the average delta(rBC)/delta(CO) ratios for that coating thickness bin.**

This is an aesthetic choice to cut-down on generally superfluous axis labelling (sometimes referred to as "chart-junk" in data visualisation discussions), so we have kept the tick marks as is.

As for colouring each coating thickness bin with the rBC:CO ratio, the data indicates that such averaging would be misleading as the ratio can vary significantly for a given coating thickness and vice-versa.

**Table 1: (L) must mean Airport. What does Phase P1 or P2 mean?**

These refer to the distinct meteorological phases following Brito et al. (2014) and Darbyshire et al. (2018) that are defined in Section 4.

We have added this additional information to the table caption.

**Anonymous Referee #2**

**The present manuscript is a highly-focused piece of work that examines chemical and microphysical property changes of biomass burning aerosol emissions that were sampled during the SAMBBA field campaign in Brazil. By and large the manuscript is written reasonably clear (explicit areas where more clarity is needed are highlighted below). As biomass burning (BB) events represent a major source of particulate matter injected into the atmosphere, it is indeed important to characterize the emissions and to understand how the chemical, microphysical, and optical properties transform as the plume ages so that models can, in turn, improve upon there fidelity. Using the analysis tool of excess mixing ratios, the authors report on a negligible net increases in organic aerosol (OA) even though there is concomitant changes in chemical properties - namely the oxidation state of the OA. This finding is inline with that reported by others and the authors put forth the same argument as that put forth by others that the loss of primary organic aerosol (POA) due to plume dilution is offset by the production of secondary organic aerosol (SOA), hence the negligible change in net OA loading. The other reported findings centers on refractory black carbon and its mixing state. In summary, the present manuscript adds important findings to a growing body of data on biomass burn-generated carbonaceous aerosol production and evolution and as such should be published. That said, to this reviewer at least, this manuscript has a feel of being incomplete in some of its analyses and in other areas left this reviewer wondering why other datasets were not part of this manuscript. This feeling could be simply that the authors are parsing out various subject matter for other manuscripts (e.g., optical properties). If this is indeed the case, the authors should be explicit about that. Once this authors address this ad some other top-level items listed below along with several specific items (e.g., figure numbering and clarification text) the manuscript should be acceptable for publication.**

**While this manuscript is highly-focused, as highlighted above, it seems that the authors miss the opportunity to role in other data sets that could, potentially, elucidate all that is**

going on in these complex emissions plumes. For example, the authors report that there are changes in the OA chemical properties and that they observe that the rBC coating thickness changes as a function of plume age, yet authors do not report on, nor reference any parallel paper, that looks at aerosol size distribution - arguably one of the most fundamental of measurements in our business. Doing so could tell us something about the roll of coagulation near the fire source as well as informing us about how the distribution evolves. Does the mode stay constant or grow? As hinted at above, perhaps the authors have the intention of publishing a separate aerosol size distribution-centric paper. If this is indeed the case they should make reference it, as the analysis of these dataset would easily complement what is being learned from the AMS. The absence of this dataset is even more puzzling given that the authors report on geometric mass mean diameter for rBC - along with estimates of the coating thicknesses. Why present microphysical properties of only one species (rBC) and not the primary particulate species (OA)?

Similarly, the authors say nothing about aerosol light scattering or light absorption. How does the scattering Angstrom exponent evolve with plume age in the near field? Does the mass scattering efficiency track what is observed with the AMS? In their rBC coating thickness analysis, the authors assume that the coating this transparent (coating refractive index of 1.5 +0i). What is the basis for this assumption? BB events are a known source of brown carbon. What does the absorption Angstrom exponent suggest? And, of course, what is the SSA doing in these first few hours where a lot of chemistry appears to be going on? As with the size distribution, perhaps the authors will report on this in a separate manuscript. It just seems to this reviewer that the absence of any reference to either the optical or size distribution datasets misses any opportunity to better examine what is going on.

In their examination of regional BB haze, the authors use the AMS tracer f44 and the ratio of rBC to CO. As the author state, rBC and CO are relative inert tracers that are strongly controlled by initial conditions, and that the ratio can provide some information about the influence of precipitation. But my question to the authors, especially when examining regional haze, is how do you know what the initial ratio was at the various sources that are contributing to the haze? Are all fires assumed to exhibit the same burning phase conditions (e.g., flaming versus smoldering)? Figure 7 indicates that the mass fraction of rBC ranged from 2% to 12.4%. Are the authors saying that the burn conditions are the same for these two bounding conditions? Here is where I would have expected some discussion of modified combustion efficiency (MCE) which might help answer this by telling us something about the initial burn conditions. Under active flaming conditions little CO is produced and more rBC while under smoldering conditions, more CO is produced and little rBC. Could this explain the variability observed in rBC mass fraction contributions or is the variability driven by differences in source fuel or subsequent cloud processing (e.g., rBC loss through precipitation)? While not listed in this paper, Darbyshire reports that a CO2 analyzer was deployed on the FAAM and thus is presumably available to use along with the CO data set, to estimate the MCE. It might b interesting to see if any of the variability in rBC mass fraction contributions can be explained by MCE. And if so, maybe the MCE can, in turn, improve the robustness of the rBC/CO ratio analysis. Finally, the authors might want to check out Collier et al., (figure 4; EST, 50, 8613, 2016) who

**reported on the relationship between OA production and MCE - where OA production was favored under smoldering conditions.**

We have not stated that rBC and CO provide a conclusive measure of the initial ratio when presented for our regional-scale observations. We have also not stated that all fires exhibit the same burning phase; past literature would suggest burn phase exists on a continuum rather than a simplified split between flaming and smouldering. We regularly state in the main text and present the detailed point-by-point measurements to illustrate the variability from flight-to-flight and within single flights/regions.

Our usage of the ratio of rBC to CO is very clear - we use it to reflect differences in air mass history that are most likely a function of both the initial fire conditions and the influence of precipitation. We feel this is a fair usage of such measurements as they do provide clues on processes that are relevant for the transformation of the aerosol burden on the regional scale.

As we noted in the introduction to our responses, we do not find that MCE is a robust measure on such scales.

**The authors state that the fire used in their case study was likely a natural fire and one that is "highly-smoldering". Do we expect a highly-smoldering fire to generate a 12.4% mass fraction of rBC?**

The reviewer appears to have confused our case study flight, B737 where the regional rBC mass fraction was 2%, with B746, which had the largest contribution of rBC with 12.4%. Regardless, Fig. 7 presents the regional boundary layer observations, which are not directly comparable with the in-plume measurements.

**The manuscript seems heavy on the figures (12 figures). The authors are encouraged to try to reduce this number.**

We are presenting a large body of observations in significant detail and feel that the manuscript benefits from the carefully chosen figures as is. Aside from Figs. 1 and 6 (the maps), each figure has at least a paragraph discussing the results and are thus central to the main manuscript text and not supplementary information.

**Specific comments**

We're unsure which exact lines the specific comments from Reviewer #2 refer to as they do not match up with the ACPD version of the manuscript – we have though identified and addressed the instances of missing figure number references. We have added our responses regarding the other aspects below.

**Abstract: page 1, Line 32: Please add "mean mass diameter" in front of "250- 290 nm".**

Amended.

**Page 4. Line 11: Please add figure number. ("1")**

Amended.

**Page 4, Line 22: Please add figure number ("1"). Also, not clear what comes after "and"**

Amended.

**Page 4, Line 30: Please add figure number ("2")**

Amended.

**Page 4, Line 35: Please add figure number ("3")**

Amended.

**Page 5, Line 4-5: How do the authors explain a slight decrease in rBC core diameter? This is a curious finding and this reviewer cannot help but wonder if the reported decrease is due to a measurement artifact as this was reported for the "case" study where the plume lifecycle could be well constrained (i.e., no cloud processing of coated rBC particles that could selectively wash out larger diameter rBC-containing particles). What were the highest number concentrations encountered near the source? Back-ofthe-envelope calculations assuming a size mode of 125 nm rBC particle and 1.5 ug/m3 suggests < 1000 particles/cc which should be low enough not to suffer from particle coincidence. Again, this is a very curious finding to simply close out a paragraph with, with no follow on statement or discussion.**

The reviewer is correct that the rBC number concentrations were low enough for particle coincidence to be relatively limited. Peak concentrations were 600 particles/cc and the estimated coincident particle concentration determined by the SP2 data processing was below 50 particles/cc.

Regarding the actual change in size, the variability is large across the length of the plume and the change is relatively minor (on the order of 10-15nm total), so we did not over-interpret our observations.

We have added a note to the paragraph on the rBC measurements to highlight the small changes in rBC parameters that may imply some change in burn conditions (which is perfectly possible over a 3-3.5 hour period), while noting that the variability is large.

**Page 5, Line 7: Please add figure number ("4")**

Amended.

**Page 5, Line 23: Please add figure number ("5") - Are both figures 4 and 5 necessary?**

Amended missing figure number. As stated previously, we feel the figures are necessary as they complement each other – based on prior presentations of our findings and existing AMS literature, including both is useful for comparative purposes.

**Page 5, Line 33: What is the useful "aging" range of the f44 marker as a tracer of age?**

Based on prior work and our own observations of the case study fire (see Fig. 3 and discussion in Section 3), we use *f44* as an indicator of the aging of the organic aerosol component (rather than aging of all particulate and gas phase material per se). As a rule of thumb, the average value for such regional scale measurements in polluted environments tends to plateau at approximately 0.2 and are interpreted as being reflective of highly aged material – higher values have been observed in remote environments e.g. see Morgan et al. (2010).

For the purposes of our manuscript, it appears that the "useful" aging range is up to approximately 0.2 as based on Figs. 9, 10 & 11, the observed values do not pass beyond this point. Thus the most aged material we observe is in this range and reflects the regional background as far as the aging of organic aerosol is concerned. We refer to this as the "end-point" of the evolution in the discussion.

**Page 6, Line 23: Please add Figure number ("5")**

Amended.

**Page 6, Lines 34-39 and page 21, Figure 8. As discussed above, perhaps examining the MCE might provide some useful insights. The delta rBC/delta CO scale in Figure 8 nominally ranges from ~0.0025 to ~ 0.02. Is this variability driven by cloud processing (i.e., precipitation) or MCE.**

Addressed previously – as stated in the text we cannot infer whether such changes are a consequence of precipitation or initial fire conditions.

**Page 25, Figure 12: Again, not to harp on the MCE theme, but it might prove interesting to examine whether the variability in coating thickness is driven by processing or MCE.**

See prior comment and our overall commentary at the beginning of our author response.

**Anonymous Referee #3**

**This paper provides an analysis of BC, OA, CO and OA oxidation state for a 2012 airborne field campaign of biomass burning emissions at Porto Velho Brazil. Data is presented in two parts, a case study of a single smoldering tropical plume and a regional analysis of 9 other regional flights. Some aspects of the paper were nicely put together, such as the observation of the evolution of smoke oxidation state. But the overall purpose of the paper on the evolution of aerosol mass is short on many important details. Trying to sort out mass evolution is quite tricky, especially for an individual plume. Accounting for the temporal evolution of the fire, controlling for combustion efficiency, linking source**

**plumes to the regional haze, all take a great deal of care. One also needs to demonstrate appropriate cross correlation between numerous parameter to ensure an apples to apples comparison to anything about temporal evolution. This is especially true in the present paper where it is clear from the regional survey work that the smoke particle properties show a lot of heterogeneity. While, I think the authors have spent a great deal of time on this paper, I found the narrative unconvincing. I think the paper probably needs significant revisions and resubmitted. At this point I think I can keep my comments to three main themes.**

1. **The paper references a great deal of "Recent activity" but the whole line of scientific thought on particle evolution came out of the ABLE, ESPRESSO, SCAR-C, SCAR-B missions of the late 80's and 1990s. These studies were much more rigorous than anything that is presented here. Liousse demonstrated he issues with particle evaporation, and Martins, Reid and Hobbs evaluated secondary production and found in well documented Lagrangian plumes samples significant production. We know the authors of this paper are aware of this work because some of the co authors were actually on these papers. A summary of this work is in the 2005 biomass burning review papers by Reid. The conclusion "secondary production is complicated and varies by fire" is indeed true, but there has been a great deal of work done in the past and even currently (all un referenced) that actually narrows down processes. Reid and Martins points are that secondary production and basic condensation happen very rapidly. Secondary production of sulfate requires cloud processing. Going back to the late 1980s significant and rapid organic acid production has been observed (I think ABLE mission). This paper lacks any concrete linkage to past knowledge to move the field forward.**

We certainly did not wish to exclude past work and relied on citations to review papers and their references in the interest of brevity. We thank the reviewer for the suggestions and have added a reference and discussion of the review paper from Reid et al. (2005) to summarise past findings.

Where we did focus on individual studies, we highlighted predominantly AMS measurements and those conducted in Brazil as these were most relevant to our paper as we relate our observations to a key AMS mass spectral marker (*f44)* which was not measured in the prior studies in the 80s and 90s.

Furthermore, past studies relied on either optical or mobility number concentration measurements that do not separate out the organic and inorganic components, thereby conflating the different processes these are subject to. Organic composition measurements were typically determined from thermal evolution techniques that are prone to biases and have far lower time-resolution than our AMS measurements. These are key aspects that differentiate our study from prior work.

We also note that our measurements differ with many of the studies noted by the reviewer in that we were primarily presenting regional measurements i.e. not "well documented Lagrangian plumes" aside from our case study. We also had different instrumental payloads, which have their pros and cons – we focussed on discussion

of processes we could more directly infer while also outlining our measurements so that future work could use our observations e.g. plume modelling to constrain the likely processes.

We feel the reviewer is making an unfair criticism of our study in that we have not claimed, nor did we set out to, quantitatively explain all the relevant processes relating to secondary aerosol production in biomass burning plumes; ours is an observational study where we have explored the various potential drivers on the transformation and aging of carbonaceous aerosol.

2. **The single case study presented is for a low combustion efficiency plume without any presented evidence that downwind samples are of the same fire characteristics. During the observation of fires in SCAR-C and SCAR B it was found that fire properties change rapidly. Given that the test case had a MCE< 0.8, then black carbon production must have been at a minimum. Perhaps there was some flaming combustion along the periphery. Therefor the relationship of secondary production to rBC is probably pretty tenuous. At the same time, most of the cases observed for secondary production have been associated with flaming combustion. Smouldering combustion is essentially a surface reaction. So with the limited data provided, I am not sure what to make of this particular test case.**

We're unsure exactly what the reviewer is referring to when they state that the "relationship of secondary production to rBC is probably pretty tenuous" – we do not focus on this as a part of our analysis of the case study – instead we relate our measurements to the distance of the smoke from the main fire and approximate age of the plume. We present the black carbon measurements as these are unique in terms of the existing literature on fire in Brazil and in general. Furthermore, they illustrate that the black carbon is rapidly coated at source, which in itself is a valuable contribution to the body of literature on black carbon in smoke plumes.

While Reviewer #3 is unsure of what to make of our particular test case, Reviewers #1 and #2 have seen the value in our observations, as well as keen interest in our findings from the wider community when this work has been presented previously. Our case study is a presentation of the fire that we were able to observe during our flight campaign and serves as a useful addition to the literature with it being on the more extreme end of smouldering fire conditions – such observations are a valuable contribution as "edge cases" are a good test of our understanding of processes when e.g. recreated in laboratory or modelling studies.

3. **Both comments one and two then project onto overall issue of making an apples to apples comparison to evaluate particle evolution. The authors report an MCE value, so there must be CO2 data available. But no time series of MCE is provided, nor even a CO to rBC ratio plot. Rather the reader has to do an eyeball comparison of the two on a log plot. For the regional samples we are presented with a great deal of variability in particle properties (other than well known oxidation with time) but not the types of additional data that other studies have used to sort out**

**what is going on. I think the authors need to spend more time on the data narrative.**

As stated previously, we do not find that MCE is a robust measure downwind or on the regional scale, and thus do not present it as it is potentially misleading.

With regards to "additional data", as discussed in the introductory commentary to our responses, we are constrained by the instrumental payload of the aircraft plus unavoidable instrument failures in what is a challenging environment to conduct measurements. There are unfortunately limits on what is possible in experimental campaigns such as these.

Given the above, the final suggestion from the reviewer that we need to "spend more time on the data narrative" is thus an invalid criticism of our study – we have spent a significant amount of time analysing the unique measurements that had good operational coverage in our study, presenting a large amount of detail and a synthesis of our findings.

---

## Author Response (AR2)

**Reviewer Comments**

We thank the editor and reviewers for their comments on our manuscript, which we have sought to answer and/or amend the revised submission. In the following we have included the reviewers' comments in bold text, with our responses beneath.

In the revised manuscript, significant additions are provided in red text.

**Editor Comments**

**1. Please define "sm" as standard meter somewhere in the paper, best in the abstract where this unit is first mentioned. It is not a traditional unit so a definition would be helpful.**

We have added a note in the method section where we defined STP and also in the abstract.

**2. Also on a subject of units, omitting space between m and s (e.g., pm page 6 line 20) produces an inverse millisecond instead of the intended meter per second. Omitting a space between ug and sm (e.g., on page 6 line 15) produces a nonsensical unit. I would search for the occurrence of these and insert spaces as needed.**

We have corrected these occurrences.

**Anonymous Referee #1**

**The overriding remaining main issue with the paper is related to asserting that the conclusions from the single case study are representative. It is unfortunate that there are no measurements of another fire plume, such as one with a higher MCE and lower initial OM to CO ratio, which could be analyzed for this paper. It could have helped to actually bridge the OM-to-CO-ratio gap between emissions and regional haze observations (asserted on P3 L26 as being bridged by this analysis), esp. the data from Flight B742 shown in Figure 8. The result is that nothing can be definitively said about the differences in OM to CO ratios in the regional haze observations and this is ultimately stated in the abstract and conclusion section. While the case study shown of one plume certainly indicates there is limited evidence of net increase in aerosol mass, this finding is presented as a generalized conclusion of the study. It is not necessary for the authors to show the representativeness of this conclusion, as long as it is clear that it arose from a single case study. Assertions that it was representative should be removed from the manuscript. In particular, the title and abstract should be revised to reduce the implication of general representativeness from the single case study analyzed.**

We have not asserted that the conclusions from our single case study are representative of the entire region. We have presented a wealth of data at the regional scale that shows limited evidence for net enhancement of OA on those spatial scales over a given flight, and we have made it clear in the text when we are discussing the case study and when we are discussing the regional-scale measurements.

Our study has sought to bridge near-source observations with broad regional-scale synthesis as we a) investigate plume aging in the near-field immediately downwind of an active fire in our case study and b) presenting detailed highly time-resolved analysis of regional-scale measurements that span a range of atmospheric aging scales i.e. not just focussed on broader assessments split by meteorology and spatial location.

The title doesn't assert anything regarding the results of the paper and is merely a descriptive statement of the focus of the paper i.e. transformation and aging of biomass burning carbonaceous aerosol.

**Minor comment related to rBC mixing state:**
**Since the last paragraph of the introduction (P3 L24) mentioned "mixing state of BC", it was expected that the fraction of coated rBC-containing particles to the total number of rBC-containing particles (in effect quantifying rBC mixing state) would have been reported, esp. as a function of plume age for the case study. Were there uncoated rBC-containing particles in the fresh emissions and did they accumulate coatings as they aged? Alternatively, if nearly all of the rBC-containing particles had measureable coatings, would a conclusion about their aging indicate that the thick coatings close to the source evaporated as they aged? At P6 L34, consider making a new paragraph about the rBC coating thickness, starting with a statement such as "nearly all rBC-containing particles had measureable coatings" or whatever the case may be. Another such statement could be added to the beginning of the paragraph discussing Figure 12 (P9 L22). Otherwise, P11 L27 appears to be the only place where the mixing state of rBC-containing particles is mentioned. The rBC mixing state information (qualitative or quantitative) could be useful to the modeling community.**

We didn't make a categorical/speculative statement on whether the reduction was due to evaporation as the change was small and the variability was large, as we noted in the main text.

We have added a sentence to the regional analysis section as per the referee's suggestion.

**Anonymous Referee #2**

**One minor revision: On page 9 lines 24-26, the authors "The broad bimodal-like structure in coating thickness in flight B737 could be linked with differences in the ratio of rBC and CO, with the thicker coatings of 80-100 nm associated with a larger ratio; conversely, the thinner coatings of less than 60 nm are coincident with the smaller ratios observed."**

**Perhaps the authors could colormap the coating histograms in Figure 12 with the delta_rBC/delta_CO ratio as they did in several other figures. It could underscore their statement in the text.**

This was suggested by Referee #1 in the APCD review phase, which we did not feel was a robust representation of the data as such averaging would be misleading as the ratio can vary significantly for a given coating thickness and vice-versa.

**Second minor revision: Page 27, Figure 8: The regression line is black not red (as written in the caption). Not sure which color is best, but the inconsistency should be corrected.**

We have updated the Figure 8 and also Figure 9 which had a similar issue.

[revised manuscript text omitted]